# Bubble-PAPR: a phase 1 clinical evaluation of the comfort and perception of a prototype powered air-purifying respirator for use by healthcare workers in an acute hospital setting

Brendan A McGrath ,[1,2] Clifford L Shelton ,[3,4] Angela Gardner,[1] Ruth Coleman,[1] James Lynch,[1] Peter G Alexander,[1,2] Glen Cooper[5]

For numbered affiliations see end of article.

**Correspondence to**
Dr Brendan A McGrath;
brendan.mcgrath@manchester.ac.uk

## ABSTRACT

**Objectives** We aimed to design and produce a low-cost, ergonomic, hood-integrated powered air-purifying respirator (Bubble-PAPR) for pandemic healthcare use, offering optimal and equitable protection to all staff. We hypothesised that participants would rate Bubble-PAPR more highly than current filtering face piece (FFP3) face mask respiratory protective equipment (RPE) in the domains of comfort, perceived safety and communication.

**Design** Rapid design and evaluation cycles occurred based on the identified user needs. We conducted diary card and focus group exercises to identify relevant tasks requiring RPE. Lab-based safety standards established against British Standard BS-EN-12941 and EU2016/425 covering materials; inward particulate leakage; breathing resistance; clean air filtration and supply; carbon dioxide elimination; exhalation means and electrical safety. Questionnaire-based usability data from participating front-line healthcare staff before (usual RPE) and after using Bubble-PAPR.

**Setting** Overseen by a trial safety committee, evaluation progressed sequentially through laboratory, simulated, low-risk, then high-risk clinical environments of a single tertiary National Health Service hospital.

**Participants** 15 staff completed diary cards and focus groups. 91 staff from a range of clinical and non-clinical roles completed the study, wearing Bubble-PAPR for a median of 45 min (IQR 30–80 (15–120)). Participants self-reported a range of heights (mean 1.7 m (SD 0.1, range 1.5–2.0)), weights (72.4 kg (16.0, 47–127)) and body mass indices (25.3 (4.7, 16.7–42.9)).

**Outcome measures** Preuse particulometer 'fit testing' and evaluation against standards by an independent biomedical engineer.
Primary: Perceived comfort (Likert scale).
Secondary: Perceived safety, communication.

**Results** Mean fit factor 16 961 (10 participants). Bubble-PAPR mean comfort score 5.64 (SD 1.55) vs usual FFP3 2.96 (1.44) (mean difference 2.68 (95% CI 2.23 to 3.14, p<0.001). Secondary outcomes, Bubble-PAPR mean (SD) versus FFP3 mean (SD), (mean difference (95% CI)) were: how safe do you feel? 6.2 (0.9) vs 5.4 (1.0), (0.73 (0.45 to 0.99)); speaking to other staff 7.5 (2.4) vs 5.1 (2.4), (2.38 (1.66 to 3.11)); heard by other staff 7.1 (2.3) vs 4.9 (2.3), (2.16 (1.45 to 2.88)); speaking to patients 7.8 (2.1) vs 4.8 (2.4), (2.99 (2.36 to 3.62)); heard by patients 7.4 (2.4) vs 4.7 (2.5), (2.7 (1.97 to 3.43)); all p<0.01.

**Conclusions** Bubble-PAPR achieved its primary purpose of keeping staff safe from airborne particulate material while improving comfort and the user experience when compared with usual FFP3 masks. The design and development of Bubble-PAPR were conducted using a careful evaluation strategy addressing key regulatory and safety steps.

**Trial registration number** NCT04681365.

## STRENGTHS AND LIMITATIONS OF THIS STUDY

⇒ We employed user-centred design, engineering optimisation and staged feasibility testing to develop a novel powered air-purifying respirator (Bubble-PAPR) for use specifically in front-line healthcare settings.

⇒ The design of Bubble-PAPR met regulatory standards and our evaluation demonstrated that it met the key requirements of comfort and perceived safety identified as essential requirements by healthcare staff.

⇒ The design and development of Bubble-PAPR were conducted using a careful strategy addressing key regulatory and safety steps, measured against UK/European standards, in contrast to many devices rapidly developed and deployed during the pandemic.

⇒ The development of Bubble-PAPR is an excellent example of growing a cosmopolitan network (social networks across historical, political and cultural boundaries).

⇒ Limitations of our study include that design and evaluation were undertaken at a single large hospital, using similar staff groups and a lack of formal independent cost analysis.

## INTRODUCTION

The COVID-19 global pandemic created a worldwide shortage of personal protective

equipment (PPE)[1] and highlighted significant usability issues in current PPE products.[2] In addition to direct contact, airborne diseases may be spread by aerosol or droplet transmission. Aerosol transmission may be mitigated by the appropriate use of respiratory protective equipment (RPE), a particular classification of PPE. However, RPE is used as part of a hierarchy of control measures. This is because RPE only protects individual workers, is prone to failure or misuse (wrong RPE for the wrong task/environment) and wearers may get a false sense of security, which may lead to neglect of other aspects of infection prevention and control, such as isolation requirements.[3] A range of inspiratory filtering devices exist: dust masks, half-face masks, full-face masks and powered (fan-assisted) respirators. Powered respirators include: half/full-face masks, helmets, hoods and visors. Though not used in healthcare, for completeness, breathing apparatuses are systems that supply an independent, positive pressure supply of breathing-quality air.

Face masks may be classified by considering the level of protection they offer the wearer to inhalation of environmental contaminants. Simple surgical face masks or 'nuisance' dust masks do not entirely filter droplets or aerosols. Filtering face piece (FFP) masks comprise layers of synthetic non-woven material with interleaved filtration layers and provide protection against small airborne particles (aerosols). Different types and constructions of FFP masks can be classified by their ability to filter small particles. Particulate filters can be classified as low (P1) to high (P3) efficiency, filtering between 80% of particles smaller than 2 µm to 99.95% of particles smaller than 0.5 µm, respectively (box 1).[4] Respiratory protection can, therefore, be considered in terms of a combination of the filtering ability of the device relative to the exposure environment and it is fit on the wearer's face. A device is considered adequate if it has the capacity to reduce the wearer's exposure to a hazardous substance to acceptable levels (to comply with occupational exposure limit values). Devices can be reusable, but the majority are single use. Masks are difficult to recycle due to their layered construction and the pandemic contributed to an unprecedented rise in RPE-related clinical waste.[5]

The majority of RPE used in healthcare settings are disposable face masks adopted from industry. Masks are not designed to be worn for long periods or repeated shifts, may restrict the visual field, limit communication, cause facial damage due to their tight fit and require multiple time-consuming 'fit tests' for each model of the device for each staff member. All these issues were highlighted in the context of the 2002–2004 SARS epidemic.[6] More appropriate solutions for prolonged and repeated use include powered air-purifying respirators (PAPRs). But, again, these are not designed primarily for healthcare, are heavy, noisy, expensive, difficult to clean to clinical standards and not suitable for the specific needs in front-line healthcare environments.

There have been several widely reported 'homemade' or 'MacGyvered' devices that well-intentioned groups or

---

**Box 1  Classification of particulate filters, with a worked example and fit testing data from EU Standard 149:2001 respiratory protective devices**

**P1**—Filters about 80% of particles smaller than 2 µm.
**P2**—Filters about 94% of particles smaller than 0.5 µm.
**P3**—Filters about 99.95% of particles smaller than 0.5 µm.

A respiratory protective device is considered adequate if it has the capacity to reduce the wearer's exposure to a hazardous substance to acceptable levels. The ratio of airborne particles outside:inside the filtering device gives a nominal (theoretical) protection factor. An assigned protection factor reflects the actual workplace conditions. For example, an airborne dust contaminant with an occupational exposure limit of 5 mg/m$^3$ may be present in the workplace in concentrations up to 60 mg/m$^3$ (determined by monitoring). A particle filter is needed to reduce the concentration by at least a factor of 12 (60/5=12). A P3 filter with an assigned protection factor of 20 would be suitable (as this is greater than the factor of 12 required). Other considerations such as exposure time, useability and disposal of the device need to be considered prior to undertaking a fit test with the intended wearer.

A fit test verifies that a specific model of device works as intended with a particular individual. For example, different face shapes and facial hair can interfere with a particular system's ability to filter environmental contaminants effectively.

Qualitative fit testing assesses the inward leakage past a mask of airborne compounds detectable by the wearer (typically bitter/sweet tasting substances), aerosolised using a spray device.

Quantitative fit testing measures particulate concentrations inside and outside of devices, typically undertaken by measuring sodium chloride aerosolised in water to generate a 'particle' count. Quantitative fit testing generates a fit factor—the ratio of airborne particle counts outside:inside. The fit factor takes account of the whole device (the filter, hood and airflow in the case of a powered air-purifying respirator (PAPR)). Fit factors for PAPRs are very high (optimal protection) and so if correctly worn, fit testing prior to use is not usually required.

---

individuals developed to protect staff and patients during the pandemic.[7] In a time of crisis, these innovations were often rapidly developed without significant funding and delivered to areas of need during a time of global RPE shortage. However, due to the urgency of the situation, few of these devices sought or achieved independent certification or provided data to support safety.[8] Turner *et al* proposed a framework for the safer adoption of novel devices,[7] which defines the problem and reviews existing solutions, benchmarks safety indices for the devices and then evaluates it in a structured manner through simulated, low-risk and then high-risk clinical settings (online supplemental table S1). Broad stakeholder feedback is encouraged through iterative review cycles, redesign and improvements.

Considering the above, our project aimed to design and produce a low-cost, ergonomic, hood-integrated PAPR for use in front-line healthcare settings. Our objectives were to focus on user-centred design, engineering optimisation, staged feasibility testing, certification, intellectual property protection and then rapid manufacture and distribution. We also aimed to design the PAPR to be reused, refurbished and recycled where possible, using

readily available, simple and interchangeable key parts which proved difficult to source during the early stages of the pandemic. Finally, by designing an available, affordable PAPR system that could be cleaned appropriately and reused between different staff, we aimed to provide equitable access to high-quality RPE that offered optimal protection to *all* staff, wherever they worked.[9] In this phase 1 clinical evaluation, we hypothesised that participants would rate Bubble-PAPR more highly than current FFP3 face mask RPE across the domains of comfort, perceived safety and communication.

## METHODS

The design team brought together front-line clinical staff based in the Wythenshawe Hospital Acute intensive care unit (ICU) of Manchester University National Health Service (NHS) Foundation Trust (MFT), an experienced product design consultancy (Designing Science, Middlesex, UK) and the technical expertise of the School of Engineering at the University of Manchester (UoM). The study protocol, analysis plan and recruitment metrics were registered and reported at ClinicalTrials.gov (NCT04681365). Participating staff were provided with participant information sheets, a detailed explanation and demonstration of the safe use of Bubble-PAPR, and written consent was obtained. User needs assessment was conducted through a series of workplace diary card exercises documenting typical activities undertaken by front-line healthcare staff, synthesised in focus groups. Staff were invited to participate (by email and posters in rest areas) from clinical locations where RPE was mandated within the hospital. The first two respondents from each area were recruited to the diary card and focus group activities. Rapid design and evaluation cycles occurred based on the identified user needs. In addition, evaluation of early prototypes occurred in simulated clinical environments, collecting usability data from participants.

### Patient and public involvement

Patient and public involvement was undertaken through the Manchester Academic Critical Care research group's patient forum. There were powerful accounts from patients who regularly described not being able to understand what hospital staff wearing PPE were saying and being troubled that they had no idea what their carers looked like. These reports led us to focus on prioritising the ease of communication with Bubble-PAPR. Staff participants who were invited to wear Bubble-PAPR were recruited from clinical locations where RPE was mandated, by direct invitation from the research team.

### Study procedures

A trial safety committee was established to oversee the results of laboratory and bench testing of the prototype, initial safety data, usability and adverse event data at each stage of the evaluation. The committee met prior to commencing clinical evaluation. It was tasked with the decision to allow the evaluation to proceed between phases: simulated clinical environment, low-risk (non-infectious) clinical environment and high-risk clinical environment (COVID-19 wards and ICUs). Early iterations of Bubble-PAPR included 3-D-printed collars and key parts (such as the impeller), along with a variety of designs of the hood. A final iteration of Bubble-PAPR included a medical-grade foam collar, precision-machined internal components and a revised (smaller) hood was further tested in high-risk environments. Prior to first use, several device safety checks were independently undertaken by the MFT Electrical and Biomedical Engineering Department and INSPEC International, Salford, UK). A short report addressing the quantitative and qualitative criteria detailed in the relevant standards, and summarised in online supplemental tables S2–S5, was presented to the trial safety committee. The first 10 study participants to wear Bubble-PAPR underwent 'fit testing' with a particulometer (TSI Portacount Fit Tester 8040, TSI Instruments, Buckinghamshire, UK) following a standard protocol derived from the UK Government's Health and Safety Executive.[10] Fit testing is not required before wearing PAPRs, including Bubble-PAPR. The purpose of fit testing was to collect device performance data and to allow the research team to assure the trial safety committee that Bubble-PAPR was performing to an appropriate standard. This INDG-479 protocol requires a 'Fit Factor' pass level of 100 for FFP3/N95 face masks and 500 for full face masks/hoods. Participants followed this standard protocol during quantitative fit testing, which involved the following exercises undertaken for at least 60 s: normal breathing, deep breathing, turning head from side-to-side, moving head up and down, talking; bending over to 90°, repeat normal breathing. European Conformity Standard EN12941 requires an applied fit factor of 40 for a 'loose-fitting hood' PAPR; the equivalent of a nominal protection factor of at least 500 (accepting an inward leakage of 0.2% with a P3 class filter see box 1). By comparison, the minimal fit factor for an FFP3 mask in a clinical environment is 100. Tests were conducted in an ICU side room with a particle generator to reach background counts between 70 000 and 100 000 particles/cm$^3$.

### Outcomes

The primary outcome was based on Davis' technology acceptance model (perceived usefulness and perceived ease-of-use overcoming barriers to adoption).[11] First, staff were asked to rate their experiences using current RPE (a variety of reuseable or disposable FFP3 masks) using a series of questions based on Likert-type scales. Next, safe use of the Bubble-PAPR was explained, and instructions for use were provided, supported by videos of donning, doffing, cleaning and storage. Bubble-PAPR was then worn during simulated/clinical use where the usual tasks were undertaken (identified in the focus groups, including verbal communication between colleagues and patients; writing; typing; reading notes, computer screens and monitors; manual handling; invasive procedures;

emergency resuscitation; airway management; and maintenance of a clean/safe bedside environment). In order to evaluate critical communication and the stability of the Bubble-PAPR, the simulated environment tests also included high-stakes team-based tasks such as managing a cardiorespiratory arrest, cardiopulmonary resuscitation, assessment and management of the critically ill patient and complex airway management. Finally, after removal (doffing) of Bubble-PAPR, staff were immediately invited to complete a second questionnaire focused on the prototype. Free-text comments were also invited.

The primary endpoint was staff rating of the comfort of Bubble-PAPR (vs current FFP3 face masks). Secondary endpoints focused on communication and perceived safety. Specifically, this was staff ratings of the prototype in terms of: how safe participants felt, ease of communication with colleagues and ease of communication with patients (again, Bubble-PAPR vs current FFP3 face masks). Additional questions explored wearer anxiety, ease of use and performance while undertaking usual work tasks. In parallel, in-house device feasibility testing was conducted in the hospital environment to test ergonomics and air particle filtration. The research framework for this study was based around in-house exemption for device development from the UK Medicines and Healthcare products Regulatory Agency. This means that the hospital, acting as manufacturer, can use a device it has developed itself internally. Such a device is not required to undergo to independent testing and therefore it will not achieve a certificate of conformity (UK-Conformity Assessed or Conformitè Europëenne marking). However, in order to assure the study sponsor and staff participants of the safety and efficacy of Bubble-PAPR, we tested against existing conformity standards for PAPRs relevant at the time of development (British Standard BS EN 12941 (Respiratory Protective Devices: Powered filtering devices incorporating a helmet or hood) and the European Union Personal Protective Equipment Directive EU2016/425).[4 12] Some of the testing was undertaken internally by independent biomedical engineers, with the flow rate and carbon dioxide testing undertaken externally.

## Sample size and statistical analysis

A pilot evaluation was conducted in August 2020 to test the questionnaires and to assess the likely population means for the test scores (online supplemental table S3). We calculated a sample size of 20 participants would be required for each phase of the evaluation to detect a significant difference between usual PPE and Bubble-PAPR, based on a mean difference of 2.5 (SD 0.9) points on the 7-point Likert scale identified during the pilot evaluation (alpha=0.05, 90% power). In addition, we allowed for a 5% drop-out and missing data rate, concluding 22 participants per phase. All variables were explored via appropriate graphical and descriptive statistics to evaluate distributions, data completeness and form. Analyses were conducted in RStudio 2020 (Boston, Massachusetts, USA, www.rstudio.com). Analyses were performed separately for each phase for presentation to the trial safety committee, with a pooled analysis conducted at the study conclusion. Comparisons between groups (current RPE vs Bubble-PAPR) were made using a paired t-test or Wilcoxon signed-rank test as appropriate.

## RESULTS

The final design of Bubble-PAPR is shown schematically in figure 1 (www.bubble-papr.com, with detailed technical drawings available by searching the patent number (PCT/GB2021/052147) at www.espacenet.com). The device safety checks and fit testing results are presented in online

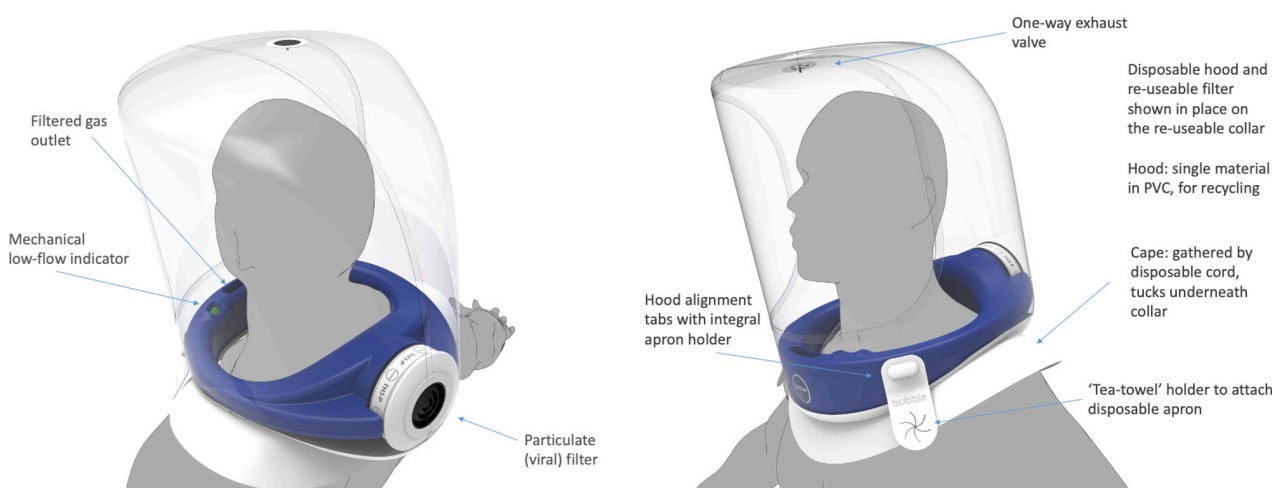

**Figure 1** Bubble-PAPR comprises a medical-grade foam neck collar and a separate polyvinyl chloride (PVC) hood. The universal fit collar draws air in through a filter via an impeller powered by an external battery. The collar has a mechanical low flow indicator and can be cleaned and reused by different users. The semirigid hood is pulled over the collar before donning and is secured by integrated straps. PAPR, National Health Service.

supplemental tables S2–S4, respectively, demonstrating a mean fit factor of 16 961. Additional particulometer tests were undertaken with deliberate tears up to 20 cm in the hood using a dummy head. The lowest fit factor recorded with the damaged hood was 1123. Therefore, the trial safety committee concluded that the Bubble-PAPR performed its primary purpose of adequately protecting staff from airborne environmental contaminants.

Fifteen staff contributed to the diary and focus group exercises. Nurses (n=7), doctors (4), physiotherapists (2), advanced practitioners (1), speech and language therapists (1) representing emergency medicine, critical care, orthopaedics and obstetric specialties generated a list of tasks to be undertaken. One staff member from the 16 invited could not attend the focus group meeting. Staff reported a range of patient-facing activities, including: verbal communication between colleagues and patients; writing; typing; reading notes, computer screens and monitors; manual handling; invasive procedures; emergency resuscitation; airway management; and maintenance of a clean/safe bedside environment. Over the course of the evaluation, staff completed all of the tasks identified by the diary exercise while wearing Bubble-PAPR in the clinical environment. Ninety-one staff wore Bubble-PAPR for a median of 45 (IQR 30–90, range 10–150) min between 3 March 2021 and 21 December 2021. All relevant staff working in relevant clinical areas were approached until a maximum of six staff had been recruited per shift (the most that the research team could reasonably accommodate per shift), or the recruitment target had been met. No staff who were approached during their clinical shifts were unwilling or unable to trial Bubble-PAPR. There were no Bubble-PAPR-related safety incidents reported during the study. Staff undertook all clinical duties identified by the focus groups and diary card exercise, either in the simulation suite (n=22) or clinical settings (n=22 low risk, n=25 high risk, n=22 high risk with final iteration). Participants predominantly declared as female (69%) and were from a range of clinical and non-clinical roles (online supplemental figure S1). Staff self-reported a range of heights (mean 1.7 m (SD 0.1, range 1.5–2.0)), weights (72.4 kg (16.0, 47–127)) and body mass indices (25.3 (4.7, 16.7–42.9)) (online supplemental figure S2). Fifty-two per cent of participants reported that they normally wore glasses, with 31% wearing glasses during the evaluation. All participants described at least 6-month experience with FFP3 face masks on a regular basis ('most shifts'), with a combination of reuseable (typically 3M 6000 Series Respirators) and single use (typically 3M Aura 9330 or equivalent) face masks. No participants described using PAPRs in the 6 months prior to recruitment. All participants completed all mandatory questionnaire sections.

With pooled data for the primary outcome, 'How comfortable do you feel in your PPE?' (Likert scale bounded by 1 (very uncomfortable) to 7 (very comfortable)), Bubble-PAPR mean score was 5.64 (SD 1.55) vs usual FFP3 face mask 2.96 (1.44; figure 2). There was a mean difference of 2.68 (95% CI 2.23 to 3.14, p<0.001). Secondary outcomes focused on communication and perceived safety. For the question, 'How safe do you feel in your PPE?', Bubble-PAPR mean score was 6.15 (0.94) vs usual FFP3 face mask 5.43 (0.98); mean difference 0.73 (95% CI 0.45 to 1.00, p<0.001; figure 2). Figure 3 demonstrates communication outcomes for all 91 comparisons of Bubble-PAPR versus usual FFP3 face masks. All adjusted comparisons were significant (p<0.001) in favour of Bubble-PAPR for communicating with both colleagues and patients (table 1 and online supplemental table S5).

Secondary outcomes where a lower Likert response was considered better are presented in online supplemental figure S3. These focused on whether staff were worried about themselves or others while wearing RPE, whether the devices caused pressure or pain or if communication was impaired. Finally, staff were asked if they had to cut short a clinical (or simulated) encounter due to

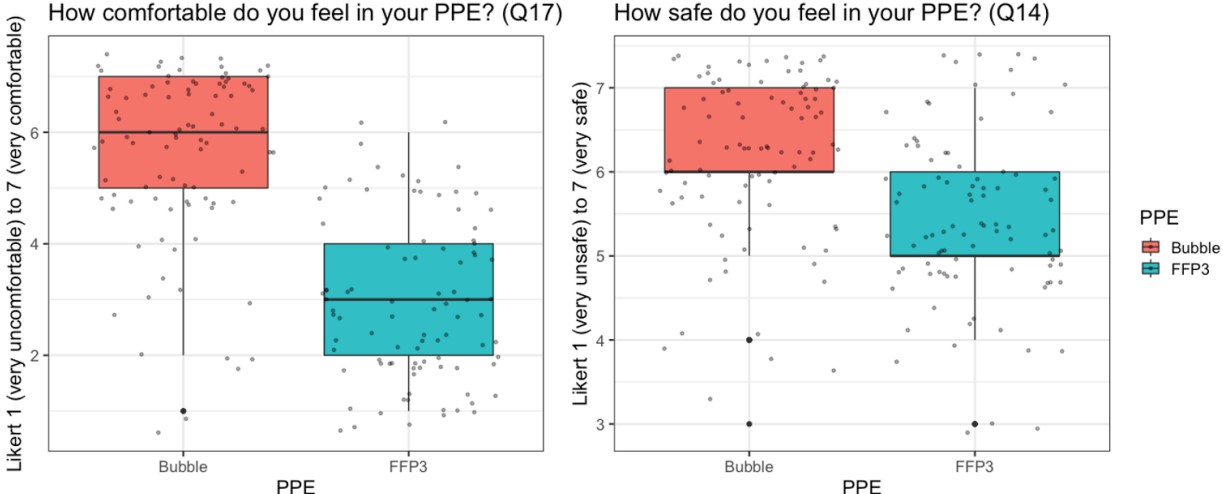

**Figure 2** Reported comfort (primary) and safety (secondary) outcomes for Bubble-PAPR versus usual FFP3 face masks. FFP, filtering face piece; PAPR, powered air-purifying respirator; PPE, personal protective equipment.

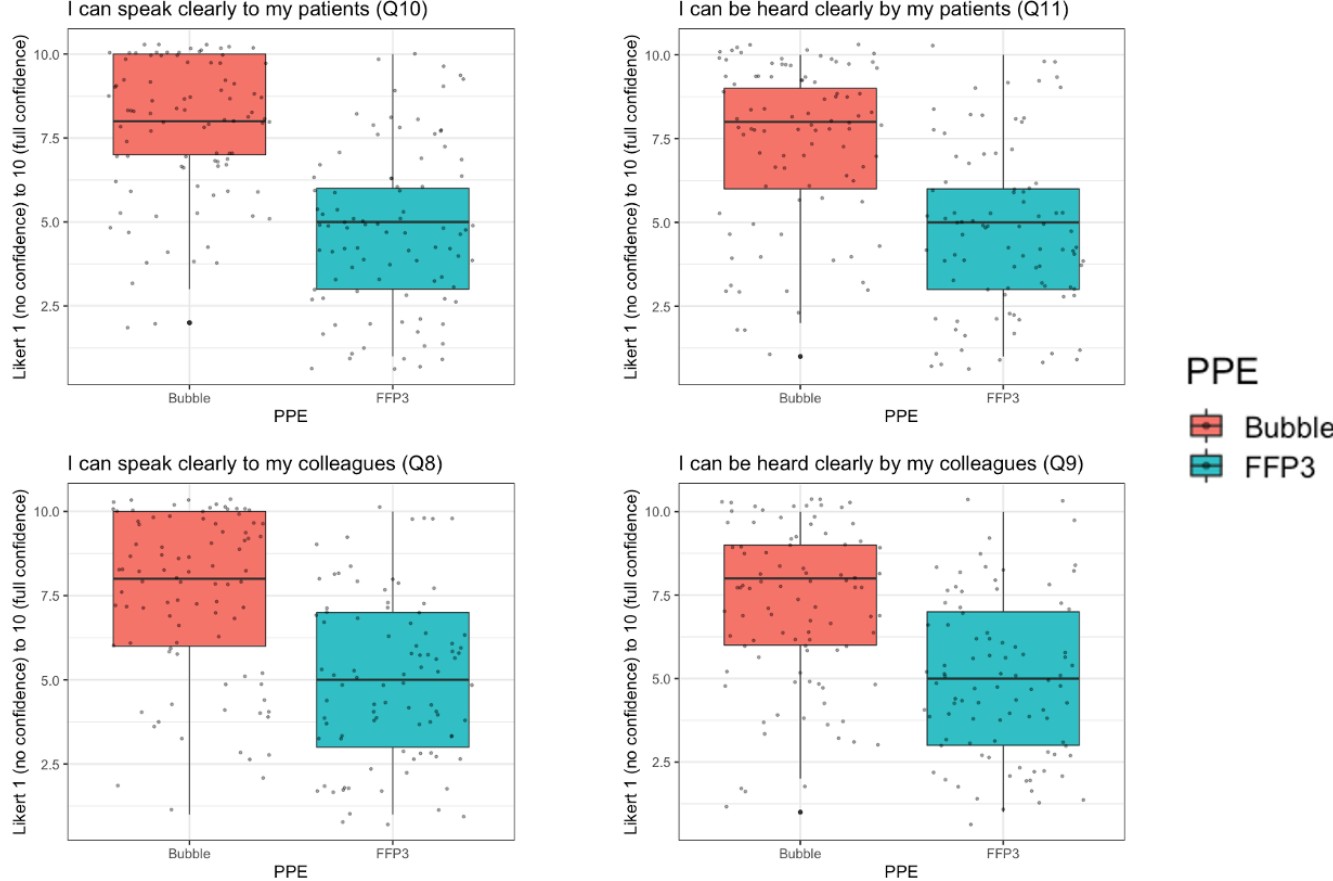

**Figure 3** Secondary communication outcomes where a higher Likert scale response was considered better. FFP, filtering face piece; PPE, personal protective equipment.

discomfort with their RPE. Again, there was a significant difference in favour of Bubble-PAPR for all metrics (all p<0.001, table 1 and online supplemental table S5).

During the initial phases, there was no significant difference between staff reporting ease of donning and doffing of Bubble-PAPR and usual PPE (the FFP3 face masks which staff had used for many months at the time of the evaluation). However, pooled results saw staff becoming more familiar with the Bubble, and Bubble-PAPR was rated easier to don and doff when compared with usual FFP3 face masks (adjusted p=0.003 and 0.002, respectively) (table 1 and online supplemental figure

**Table 1** Rating scales, summary results and comparisons across the primary outcome questionnaire domains

| | PPE | Q8 speak to staff | Q9 be heard by staff | Q10 speak to patient | Q11 be heard by patient | Q14 how safe does it feel | Q17 comfortable |
|---|---|---|---|---|---|---|---|
| Rating scale | From: | 0—no confidence | 0—no confidence | 0—no confidence | 0—no confidence | 1—very unsafe | 1—very uncomfortable |
| | To: | 10—fully confident | 10—fully confident | 10—fully confident | 10—fully confident | 7—very safe | 7—very comfortable |
| RPE type | FFP3 | 5.1 (2.4) (1–10) | 4.9 (2.3) (1–10) | 4.8 (2.4) (1–10) | 4.7 (2.5) (1–10) | 5.4 (1.0) (3–7) | 3 (1.4) (1–6) |
| | Bubble | 7.5 (2.4) (1–10) | 7.1 (2.3) (1–10) | 7.8 (2.1) (2–10) | 7.4 (2.4) (1–10) | 6.2 (0.9) (3–7) | 5.6 (1.6) (1–7) |
| Comparison | Mean difference | 2.38 | 2.16 | 2.99 | 2.7 | 0.73 | 2.68 |
| | 95% CI | 1.66 to 3.11 | 1.45 to 2.88 | 2.36 to 3.62 | 1.97 to 3.43 | 0.45 to 0.99 | 2.23 to 3.14 |
| | | Favours Bubble | Favours Bubble | Favours Bubble | Favours Bubble | Favours Bubble | Favours Bubble |
| | Adjusted p value | <0.001 | <0.001 | <0.001 | <0.001 | <0.001 | <0.001 |

FFP, filtering face piece; RPE, respiratory protective equipment.

S4). One hundred and thirty-two additional free-text comments were reviewed and categorised into positive (n=47, 35.6%), negative (67, 50.8%) and neutral (18, 13.6%) comments (online supplemental figures S5–S7). Most comments focused on the noise of the device, which improved throughout the project as the impeller and motor were made quieter in later design iterations. The categories and nature of comments were as follows: noise (33 comments (3 neutral, 30 negative)), comfort (24 comments (20 positive, 2 neutral, 2 negative)), communication (22 comments (5 positive, 6 neutral, 11 negative)), general (21 comments (17 positive, 2 neutral, 2 negative)), vision (14 comments (1 positive, 4 neutral, 9 negative)), wear and fit (10 comments (2 positive, 1 neutral, 7 negative), stethoscope (5 negative comments), safety (2 positive comments) and battery (1 negative comment).

## DISCUSSION

Our project developed an innovative prototype PAPR explicitly designed for prolonged healthcare use in high-risk clinical environments. Bubble-PAPR achieved its primary purpose of protecting staff by exceeding recognised safety standards for PAPRs, while also being rated significantly higher for comfort (the primary outcome), perceived safety, and communication with colleagues and patients (secondary outcomes) than usual FFP3 face masks. Bubble-PAPR was used in all relevant simulated and clinical scenarios identified by detailed staff diary cards, making the results of this study extremely relevant to hospital-based healthcare workers.

Bubble-PAPR was rapidly developed based on the lived experiences of front-line staff during the early stages of the COVID-19 pandemic, addressing the unmet needs of reliable, high-quality, universal and available RPE with improved comfort and communication when compared with usual FFP3 face masks. Staff overwhelmingly recognised the importance of facial visualisation when communicating with colleagues and patients. When combined with the improved comfort of wearing a PAPR over usual RPE, participants rated Bubble-PAPR consistently highly across all comparator domains.

This relatively simple evaluation study was preceded by a rapid design and prototyping phase, producing a working prototype within a few weeks. Despite the speed and agility demonstrated by the design team, we adhered to relevant conformity standards for PAPRs, following a tiered evaluation within the governance structure of an approved and regulated research project. Bubble-PAPR was only introduced into higher-risk environments following review by the trial safety committee. This structured approach contrasted with some other rapidly developed or adopted pandemic RPE systems.[7 13 14] While the PPE shortages experienced during the pandemic drove many of these innovations and adaptations, we recognised the importance of a methodical approach to design, development and testing of our prototype, both in the laboratory and

clinical settings. We recommend others to follow the framework proposed by Turner *et al* for the development of novel medical devices, with regular reviews of safety and useability data within the framework of a robust and transparent clinical trial.[7] The development of Bubble-PAPR required the rapid formation of a cosmopolitan network of front-line healthcare staff, designers, engineers, academics, innovators, marketing experts, manufacturers and funders. Our collaborative had not all worked together before and members crossed historical, political and cultural boundaries to work effectively together. Postpandemic, cosmopolitan networks such as this could become a key feature of future system resilience and facilitate new ways of working.

Our study has some limitations. Some of the endpoints were self-reported by participating staff and not independently verified. This included communication between colleagues, and between staff and patients. However, staff were performing their usual clinical duties while wearing Bubble-PAPR and any limitations of two-way communication were recognised and reported. The design of Bubble-PAPR addressed many of the issues identified by the same staff who subsequently evaluated the prototype. While our study protocol allowed evaluation only within our trust owing to the 'in-house' manufacturing exemption for testing, it is not unreasonable to expect similar results if our prototype were evaluated elsewhere. Although this may be considered a weakness of the study, many of the shortcomings of the PPE provided to front-line health workers around the world are well described and are essentially the same as those identified in our project.[15 16] Furthermore, we evaluated Bubble-PAPR against single-use and reusable FFP3 face masks, which could be construed as comparing two different classes of RPE. However, Bubble-PAPR was designed and developed to provide a viable alternative to FFP3 class face masks, in contrast to the more usual healthcare use of PAPRs. Other PAPRs are more complex, more cumbersome (belt-worn fans and hoses), more costly, and typically are selectively available on a limited basis to specific users or groups because of these factors. Although a pricing structure is currently unavailable, the simplicity of the design and components (designed with pandemic supply chain limitations in mind) means that Bubble-PAPR is likely to cost around 25%–50% of the list price of equivalent PAPRs. Our detailed analysis of work diary cards from various clinical staff ensured that Bubble-PAPR was used for all relevant procedures identified by participating staff in our settings that were undertaken by medical, nursing, healthcare assistant, allied healthcare professional (speech and language therapy, physiotherapy, pharmacy), administrative and domestic staff in the clinical area. Staff were able to undertake their usual duties with this simple, collar-worn PAPR. Limitations of the design include the inability to use a

conventional stethoscope (although Bluetooth stethoscopes were used effectively), potential visual distortions if the visor section of the hood became creased, and the residual noise during use (common among PAPRs). Although the design is simple, with visual/mechanical indicators instead of electronic indicators or alarms, this did not impact on conformity testing or function. Addressing the actual activities undertaken by specific staff groups, testing safety, performance and the user experience, is unique within published RPE product evaluation studies.[17 18] High acuity activities such as CPR and tracheal intubation were undertaken while wearing Bubble-PAPR but we collected data only around perceived comfort, safety and self-reported efficacy. Bubble-PAPR meets current industrial standards for the safe use of respiratory protection, but such standards are not usually designed with healthcare procedures in mind. Postpandemic conformity requirements will vary around the world and future iterations of Bubble-PAPR may need to adapt to meet country-specific requirements.

Our study did not directly evaluate the patient experience with staff wearing different RPE. However, the patient experience was reflected in the user specifications identified around communication, and anecdotal feedback was positive from patients, especially around facial visibility and verbal and non-verbal communication. In addition, when contrasted with FFP3 face masks, speech and language therapists reported that demonstrating speech and swallow exercises was suddenly possible with Bubble-PAPR and that the transparent nature of the hood overcame the communication barriers that can be so devastating for those with hearing impairments.[19] Although designed to be potentially recyclable, future work should address the environmental impact of polyvinyl chloride (PVC) hoods with reusable collars compared with single-use or reusable FFP3 face masks.

## CONCLUSIONS

Our study has demonstrated that Bubble-PAPR achieved its primary purpose of keeping staff safe from airborne particulate material while improving comfort, communication and the user experience when compared with usual FFP3 face masks worn throughout the pandemic. It is likely that the patient experience was also enhanced. Bubble-PAPR has been patented (PCT/GB2021/052147) and subsequently licensed to a UK-based healthcare manufacturer for large-scale manufacture and distribution to front-line NHS and other workers. The pandemic drove unprecedented collaboration between clinicians, academics and industry. The development of Bubble-PAPR is an excellent example of growing a cosmopolitan network across historical, political and cultural boundaries that could become a key feature of future system resilience.

**Author affiliations**
¹Acute Intensive Care Unit, Wythenshawe Hospital, Manchester University NHS Foundation Trust, Manchester, UK
²Manchester Academic Critical Care, Division of Infection, Immunity and Respiratory Medicine, Manchester Academic Health Science Centre, Manchester, UK
³Department of Anaesthesia, Wythenshawe Hospital, Manchester University NHS Foundation Trust, Manchester, UK
⁴Lancaster Medical School, Lancaster University, Lancaster, UK
⁵The University of Manchester School of Mechanical Aerospace and Civil Engineering, Manchester, UK

**Acknowledgements** We are grateful to our funders (detailed below) for supporting this project and the staff who participated in the evaluation. In addition, we are indebted to the designers, engineers and staff who gave their time freely during the early stages of the COVID-19 pandemic to work tirelessly on designing, building, testing and refining Bubble-PAPR. Specifically: Patrick Hall, Designing Science Ltd (www.designingscience.co.uk); Andrew Spragg, Industrial design consultant; Andrew Forbes, XK Design Ltd; James Corden, Manchester University Hospital NHS Foundation Trust Innovation Team; Nick Duggan, Innovation Consultant, Zuas (www.zuas.io); and GAMA Healthcare, Hemel Hempstead, UK (www.gamahealthcare.com).

**Contributors** All authors critically revised the manuscript for important intellectual content and approved the final manuscript. BAM attests that all listed authors meet authorship criteria and that no others meeting the criteria have been omitted. BAM acts as guarantor. BAM: conception and design, collection, analysis and interpretation of data, drafting and revision of the manuscript, and final approval of the version to be published. Participant recruitment. CLS: qualitative work package conception and design, analysis and interpretation of data, participant recruitment, drafting and revision of the final manuscript, and final approval of the version to be published. AG: qualitative work package design, analysis and interpretation of data. Drafting and revision of the final manuscript, and final approval of the version to be published. RC: design, collection and interpretation of data, drafting and revision of the manuscript, and final approval of the version to be published. JL: design, collection and interpretation of data, drafting and revision of the manuscript, and final approval of the version to be published. PGA: design, collection and interpretation of data, drafting and revision of the manuscript, and final approval of the version to be published. GC: design, collection, and interpretation of data, drafting and revision of the manuscript, and final approval of the version to be published, manufacturing and engineering lead.

**Funding** This project was supported by unrestricted grants and funding from Engineering and Physical Sciences Research Council (EPSRC) Impact Acceleration Account 302, Oxford Road Corridor (grant award number N/A), Health Innovation Manchester 'Momentum' special projects fund 2021 (grant award number N/A), Acute ICU Charitable Research Fund, Manchester University NHS Foundation Trust (grant award number N/A) and Manchester University NHS Foundation Trust (grant award number N/A).

**Competing interests** Manchester University NHS Foundation Trust, the University of Manchester and Designing Science Ltd have agreed commercial terms to license Bubble-PAPR for manufacture and development. No other competing interests are declared.

**Patient and public involvement** Patients and/or the public were involved in the design, or conduct, or reporting, or dissemination plans of this research. Refer to the Methods section for further details.

**Patient consent for publication** Not applicable.

**Ethics approval** This study involves human participants and Research Ethical and Health Research Authority approval (IRAS ID:288493, REC Ref:21/WA/0018) was granted from Wales REC5 on 27 January 2021. Participants gave informed consent to participate in the study before taking part.

**Provenance and peer review** Not commissioned; externally peer reviewed.

**Data availability statement** Data are available on reasonable request. Due to the commercial sensitivity of the intellectual property licensed at the conclusion of this project, the full dataset is not publicly available. However, the corresponding author will consider requests to disclose the dataset on an individual basis if necessary.

**ORCID iDs**
Brendan A McGrath http://orcid.org/0000-0003-3048-3480
Clifford L Shelton http://orcid.org/0000-0002-8438-398X

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
