## [Reviewer comments · BMJ Open]

ARTICLE DETAILS

TITLE (PROVISIONAL)	Bubble-PAPR: a phase 1 clinical evaluation of the comfort and perception of a prototype powered air-purifying respirator for use by healthcare workers in an acute hospital setting
AUTHORS	McGrath, Brendan; Shelton, Clifford; Gardner, Angela; Coleman, Ruth; Lynch, James; Alexander, Peter G; Cooper, Glen

VERSION 1 - REVIEW

REVIEWER	Gilbert, Catherine Tulane University School of Medicine
REVIEW RETURNED	27-Sep-2022

GENERAL COMMENTS	Abstract: 2 primary objectives are listed: 1) PAPR that provides optimal and equitable protection to all staff and, 2) that the PAPR would be rated more highly than current FFP3 face masks. To fulfill the first objective, you must demonstrate safety and efficacy of the PAPR design. Therefore, in the: Design: - Detail which specific lab-based safety standards were tested and established against the BS-EN-12941 and EU2016/425 Outcome measures: - Detail how the lab-based tests were performed and what specifics were measured. Results: - Share the most critical results that demonstrate safety and efficacy of the PAPR device. Trial Registration: - I see that data has been submitted to clinical trials.gov but I believe the public is still not able to see it. Does that mean it is still under review/revision? Strengths and limitations - What specifically makes your design best for the frontline healthcare setting? - What specific regulatory and safety steps were taken to ensure the safety and efficacy of the device? - What are the limitations of your study? There should not only be strengths listed. o No data or design of the PAPR is demonstrated o The primary participants of the study were also instrumental to the design development of the device Introduction - I would recommend changing the tone of the 4th paragraph of the introduction. The homemade "MacGyvered" devices were not only well intentioned but were done with minimal
---

funding and were developed and distributed in a short period of time to provide PPE options for those who had none in a global emergency. It is unfair to directly compare this current device with its sponsorship and longer time frame to those devices without caveats and clarification.

- Given table 1, what qualitative and quantitative fit testing did you do for your safety indices?

Methods

- What were the “typical daily activities” that this PAPR was designed for based on the diary cards?

- In the 3rd paragraph of the methods section you indicate that there have been several iterations of the device: please describe the problems that arose from each iteration and how you addressed them.

- What were the movements performed during the particulometer test?

- What other tests were performed to determine the safety and efficacy of the PAPR design? Please describe.

- What were the questions used to rate the user’s experience for both the FFP3 and the PAPR? Perhaps a table or figure?

- In the 5th paragraph you mention that you tested “ergonomics”, what does this mean? What was the test performed and how?

- What conformity standards did you test against for the Buble PAPR? What made them relevant and why are the tests performed sufficient to determine safety and efficacy of the PAPR? The reason demonstrating the safety and efficacy of the Bubble-PAPR matters to this paper that focuses on the preference and advantages of this PAPR over FFP3 masks is that, if further iterations of the design need to be made in order to achieve safety and efficacy (higher air flow rate, repositioning of components, etc) it might affect the ergonomics and usability of the design, making it either more or less user friendly, thus changing the way users rate the device.

- What do you mean by “all variables were explored via appropriate graphical and descriptive statistics”? Can you describe this please?

Results

- Please provide an actual description of the general design of the Buble-PAPR and provide the results of the tests performed for the PAPR conformity standards for design safety and efficacy.

- What are the lists of tasks/clinical duties gathered from the diary cards?

- What are the communication outcomes that were tested and how were they tested? The figure shows “I can be heard clearly by patients/colleagues”. Was this verified in any capacity? If not, please note this in the discussion as a limitation of the study as we cannot be sure that the other party was able to hear the wearer.

- Please report percentages of people who made certain categories of negative/positive comments. (i.e. 45% of people noted they had difficulty hearing wearing the PAPR, 65% commented they could see faces better, etc.). The word clouds do not give a quantitative summary of the results

- You mention that different iterations of the design made the impellor and motor quieter. Were safety and efficacy tests re-run on this iteration? Were any participants given the new iteration to wear with their daily activities and redo the questionnaire? If not, why? And please list a limitation in discussion.

	Discussion  - Please make sure you publish data in your methods and results section to support you claim that the “Bubble-PAPR achieved its primary purpose of protecting staff from airborne potentially infectious material”. You must provide evidence. - What are the “relevant simulated and clinical scenarios” that were determined by the diary cards? Were safety and efficacy tests done while performing these tasks? Why not? - In the 3rd paragraph you say, “we adhered to relevant standards following a tiered evaluation within the governance structure”. Please detail what these standards were and how it was evaluated in the Methods and Results sections. - Please explain or better support the sentence that starts, “this approach contrasted with many rapidly developed...”. It comes off as very dismissive without much specific evidence to support your claim. To what devices are you referring? What made them so reckless and inappropriate? Why is yours so much better as you do not currently share any data demonstrating safety or efficacy of your design? - How is the Bubble PAPR different from other PAPRs? - How does this PAPR’s design allow for these daily activities noted in the diary cards? What are the limitations of the design of this PAPR? - Please expand upon the last sentence of the 4th paragraph. What specifically is unique about your study and how did you test it? What did the other studies not address? Conclusion  - First sentence: safety and efficacy must be demonstrated in this paper before this statement can be made. - If the design is patented, there should be no issue discussing the design and why it is superior and more effective than other designs. Please do so in the discussion. There also should be no problem publishing data to support its safety and efficacy. - I recommend changing the last sentence. The last sentiment of your paper should not be putting other designs or studies down.
--	--

REVIEWER	Kothakonda , Akshay Massachusetts Institute of Technology, Aeronautics and Astronautics
REVIEW RETURNED	31-Oct-2022

GENERAL COMMENTS	 1. On Introduction section and Table 1, High Efficiency filter is defined as filtration efficiency of 99.95% of particulates smaller than 0.5 microns. What standard is this definition from? NIOSH defines HE filters as having 99.97% efficiency for particulate size of 0.3 micron. 2. On table 1, calculation of APF for a P3 filter is not clear. If the filtration efficiency is 99.95%, PF should be 2000. How does this relate to APF of 20 for this filter? A more thorough definition of these terms may make it clearer. Also, a definition of qualitative fit factor it’s distinction from PF and APF would be helpful to a reader, particularly in the discussion on the Methods section, relating nominal protection factor to applied fit factor. 3. In the Introduction section, the authors correctly point out how many home-made PPE devices made during the pandemic lacked proper engineering design and safety evaluations. However, there
--

	have been some efforts, particularly from small industry and academia that have carried out PPE and PAPR development adhering to thorough safety design principles for these devices. For a complete background pertaining to this work, it might be worth highlighting some of these efforts in the Introduction section. 4. One of the limitations of the current masks such as N95 masks for its use in clinical setting is the need for fit tests, as it has been pointed out in the paper. However, the subjects in the study also had to undergo qualitative fit tests prior to clinical use. This might be a limitation of the device. If the device is made so that no qualitative fit test is needed, then tests must be done (or results from tests already done) to show how many people failed fit tests. If these results don't exist, or if further tests cannot be undertaken, that should be noted as a limitation. 5. In the "Strengths and limitations of this study" section, only strengths are mentioned. Please mention limitations as well. 6. Abstract mentions that one of the objectives is to develop a low cost PAPR. Note the cost of Bubble PAPR so as to justify the accomplishment of this objective.
--	---

VERSION 1 – AUTHOR RESPONSE

We thank both reviewers and the editor's for their time taken to provide thoughtful reviews of our paper. We have attempted to address their comments in turn below. We have summarised changes to the relevant sections of the manuscript in line to hopefully make the review process simpler. We have added our responses in blue, and the *manuscript sections in italics with grey highlights* (with *blue italics* representing new text). We are pleased to present a tracked changes manuscript and a 'clean' version also.

Reviewer: 1 Dr. Catherine Gilbert, Tulane University School of Medicine

Comments to the Author:

Your design and goal are laudable and would be a benefit to the literature. However, I do believe you should publish more of the data and design in order to demonstrate the efficacy and validity of your PAPR. Please see the attached document for a more detailed review.

We thank reviewer 1 for their comments and are glad that the overall impression that our paper makes is positive. We have attempted to address your helpful comments below.

Abstract:

2 primary objectives are listed:

- 1) PAPR that provides optimal and equitable protection to all staff and,
- 2) that the PAPR would be rated more highly than current FFP3 face masks.

To fulfill the first objective, you must demonstrate safety and efficacy of the PAPR design.

Therefore, in the:

Design:

- Detail which specific lab-based safety standards were tested and established against the BS-EN-12941 and EU2016/425

The abstract word count limits our response in this section. The British and European standards for PAPRs cover: materials; inward particulate leakage; breathing resistance; clean air filtration and supply; carbon dioxide elimination; exhalation means; and electrical safety. We detail the exact standards against which the Bubble was tested against in Table 3 (supplemental). This section now reads:

Lab-based safety standards established against British Standard BS-EN-12941 and EU2016/425 covering materials; inward particulate leakage; breathing resistance; clean air filtration and supply; carbon dioxide elimination; exhalation means; and electrical safety.

Outcome measures:

- Detail how the lab-based tests were performed and what specifics were measured.

Again, we are limited by word count in the abstract. We have expanded on your suggestion in the main methods section of the paper. The actual data from the pre-use particulometer testing is detailed in Table 5 (supplemental). The abstract section now reads:

Outcome measures: Pre-use particulometer “fit testing” and evaluation against standards by independent biomedical engineer. Primary: perceived comfort (Likert scale) Secondary: perceived safety, communication.

Results:

- Share the most critical results that demonstrate safety and efficacy of the PAPR device.

This section now reads:

Results: Mean fit factor 16,961 (ten participants). Bubble-PAPR mean comfort score 5.64(SD 1.55) versus usual FFP3 2.96(1.44) (mean difference 2.68 (95% CI 2.23-3.14, $p < 0.001$)). There was a significant difference in favour of Bubble-PAPR across all secondary outcomes.

Trial Registration:

- I see that data has been submitted to clinical trials.gov but I believe the public is still not able to see it. Does that mean it is still under review/revision?

The results have been provided to the NLM but the reviewer is correct in that the public view of the results is not available. The results were submitted by the Sponsor (MFT) on Feb 15th 2022, with a revision submitted on 22nd Feb 2022 after addressing queries from the NLM team. We will check with the Sponsorship team why the results are not public. Thank you for highlighting this. The trial protocols and objectives are detailed and are publicly viewable.

Strengths and limitations

- What specifically makes your design best for the frontline healthcare setting?
- What specific regulatory and safety steps were taken to ensure the safety and efficacy of the device?
- What are the limitations of your study? There should not only be strengths listed.
 - o No data or design of the PAPR is demonstrated
 - o The primary participants of the study were also instrumental to the design development of the device

For the 'strengths and weaknesses' section, we are again limited by word count. We accept the reviewer's helpful suggestions, however. We have expanded our discussion to reflect on the design.

Figure 1 details the key design elements of the Bubble. We provide the patent number in the conclusion section which allows interested readers to access the detailed design. We feel that for a clinical journal with predominantly clinical readership, the design details are adequately covered, with links to the detailed content as required.

We have added the following statements to the 'strengths and limitations' section:

- *The design of Bubble-PAPR met regulatory standards and our evaluation demonstrated that it met the key requirements of comfort and perceived safety identified as essential requirements by healthcare staff.*
- *The design and development of Bubble-PAPR were conducted using a careful strategy addressing key regulatory and safety steps, measured against UK/European standards, in contrast to many devices rapidly developed and deployed during the pandemic.*
- *A limitation of our study was the design and evaluation were undertaken at a single (large) hospital, using similar staff groups (but different staff).*

Main paper

Introduction

- I would recommend changing the tone of the 4th paragraph of the introduction. The homemade "MacGyvered" devices were not only well intentioned but were done with minimal funding and were developed and distributed in a short period of time to provide PPE options for those who had none in a global emergency. It is unfair to directly compare this current device with its sponsorship and longer time frame to those devices without caveats and clarification.

Thank you for this insightful comment. We have reflected on this paragraph, which now reads:

There have been several widely reported 'homemade' or 'MacGyvered' devices that well-intentioned groups or individuals developed to protect staff and patients during the pandemic.⁷ In a time of crisis, these innovations were often rapidly developed without significant funding and delivered to areas of need during a time of global RPE shortage. However, due to the urgency of the situation, none of these devices sought or achieved independent certification or provided data to support safety.⁸

- Given table 1, what qualitative and quantitative fit testing did you do for your safety indices?

The third paragraph of the methods details our testing, with the results presented in Tables 3 and 5 (supplemental). We have amended this paragraph to cover your comments, which now reads:

A short report-addressing the qualitative and qualitative criteria detailed in the relevant standards, and summarised in Table S2, S3 and S5 (Supplemental), was presented to the Trial Safety Committee.

Methods

- What were the "typical daily activities" that this PAPR was designed for based on the diary

Cards?

Thank you for highlighting this oversight. We have assumed that the reader will understand what typical activities frontline staff undertake in the ICU, but appreciate that we should explain this. These activities are now described in the results section. The second paragraph now reads:

Fifteen staff contributed to the diary and focus group exercises, generating a list of tasks to be undertaken. One staff member from the 16 invited could not attend the focus group meeting. Staff reported a range of patient-facing activities, including: verbal communication between colleagues and patients; writing; typing; reading notes, computer screens and monitors; manual handling; invasive procedures; emergency resuscitation; airway management; and maintenance of a clean/safe bedside environment.

- In the 3rd paragraph of the methods section you indicate that there have been several iterations of the device: please describe the problems that arose from each iteration and how you addressed them.

We had tried to limit the design detail we presented in the paper, as discussed above. However, in response to your comments, we have added the following text to paragraph 3 of methods, which now reads:

Early iterations of Bubble-PAPR included 3-D-printed collars and key parts (such as the impellor), along with a variety of designs of the hood. A final iteration of Bubble-PAPR included a medical-grade foam collar, precision-machined internal components and a revised (smaller) hood was further tested in high-risk environments.

- What were the movements performed during the particulometer test?

We are happy to provide additional detail of the INDG-479 protocol. We have added the following text:

Participants followed this standard protocol during quantitative fit testing which involved the following exercises undertaken for at least 60 seconds: normal breathing; deep breathing; turning head from side-to-side; moving head up and down; talking; bending over to 90 degrees; repeat normal breathing.

- What other tests were performed to determine the safety and efficacy of the PAPR design?

Please describe.

All of the safety and efficacy tests that were undertaken are detailed in Tables S3 and S5 (supplemental), in addition to the user feedback metrics presented in full in Table 6.

- What were the questions used to rate the user's experience for both the FFP3 and the PAPR? Perhaps a table or figure?

All of the questions we asked are presented in full in the supplemental material (pages 27-36 of the original submission).

- In the 5th paragraph you mention that you tested "ergonomics", what does this mean? What was the test performed and how?

These are detailed in Table S3 (supplemental). Specifically, the rows cover the standard, the test detail and include any relevant notes. We have added a screen shot of Table S3 for convenience below. This details the opinion of the independent biomedical engineer and infection prevention and control staff when considering:

- Suitable resistance to wear and tear
- No sharp edges
- Fits a range of head sizes
- Does not distort vision
- Permits an appropriate field of view

supplemented by the qualitative or quantitative data collected in the questionnaires. Standards used were British Standard EN12941 (BS, 2008) and the European Regulations for Respiratory Protective Equipment EU2016/425 (ER, 2016).

Relevant section of standard	Standard detail	Test location	Test detail	Results/notes	Pass/Fail
BS 6.1.1 ER 1.3.2	Suitable resistance to wear and tear	MFT	PAPR units inspected after 1 week of continual use. Images taken before and after.	Opinion. Baseline inspection +/- photograph. Review after 1 week	Pass 14/3/21
BS 6.1.4 ER 1.2.1.2	No sharp edges	MFT	Visual and physical inspection Reports from staff evaluation	Opinion. Baseline inspection +/- photograph. Review after 1 week	Pass 14/3/21
BS 6.3.2	Fits a range of head sizes	MFT	Ten participants will undergo fit testing. These participants will have height, weight and head circumference measured as part of this standard process.	Fit test data shared with EBME team. All fit factors >500 as per BS EN 12941 standard.	Pass 25/2/21
BS 6.3.3.1	Does not distort vision	MFT	The optical area appears transparent.	Inspection by EBME team	Pass 25/2/21
BS 6.3.3.2	Permits appropriate field of view	MFT	Reports from staff evaluation	Review of results from initial staff evaluations	Pass 14/3/21

- What conformity standards did you test against for the Bubble PAPR? What made them relevant and why are the tests performed sufficient to determine safety and efficacy of the PAPR? The reason demonstrating the safety and efficacy of the Bubble-PAPR matters to this paper that focuses on the preference and advantages of this PAPR over FFP3 masks is that, if further iterations of the design need to be made in order to achieve safety and efficacy (higher air flow rate, repositioning of components, etc) it might affect the ergonomics and usability of the design, making it either more or less user friendly, thus changing the way users rate the device.

The penultimate paragraph of the methods details the two standards against which Bubble-PAPR was tested (with references):

We tested against existing conformity standards for PAPRs relevant at the time of development (British Standard BS EN 12941 [Respiratory Protective Devices: Powered filtering devices incorporating a helmet or hood] and the European Union Personal Protective Equipment Directive EU2016/425).^{4 11}

These are the standards which all PAPRs in the UK/Europe needed to meet, as determined by an independent assessor. The reviewer is correct that any further iterations of the design will require re-testing of the device against these standards.

We agree that we could have made our study methods clearer and so have amended the penultimate paragraph of the methods section to now read:

The research framework for this study was based around in-house exemption for device development from the UK Medicines and Healthcare products Regulatory Agency (MHRA). This means that the hospital, acting as manufacturer, can use a device it has developed itself internally. Such a device is not required to undergo to independent testing and therefore it will not achieve a certificate of conformity (UK-Conformity Assessed or Conformité Européenne marking). However, in order to assure the study Sponsor and staff participants of the safety and efficacy of Bubble-PAPR, we tested against existing

conformity standards for PAPRs relevant at the time of development (British Standard BS EN 12941 [Respiratory Protective Devices: Powered filtering devices incorporating a helmet or hood] and the European Union Personal Protective Equipment Directive EU2016/425).^{4 11} Some of the testing was undertaken internally by independent biomedical engineers, with the flowrate and carbon dioxide testing undertaken externally.

- What do you mean by “all variables were explored via appropriate graphical and descriptive statistics”? Can you describe this please?

This is a standard way of examining the raw data to establish if it is normally or non-parametrically distributed or skewed. This then informs the correct statistical analysis. We have also looked for missing data. We have re-worded the section as below, which we hope makes our intentions clearer:

All variables were explored via appropriate graphical and descriptive statistics to evaluate distributions, data completeness and form.

Results

- Please provide an actual description of the general design of the Bubble-PAPR and provide the results of the tests performed for the PAPR conformity standards for design safety and efficacy.

As above; Figure 1 details the key design elements of the Bubble. We provide the patent number in the conclusion section which allows interested readers to access the detailed design. We feel that for a clinical journal with predominantly clinical readership, the design details are adequately covered, with links to the detailed content as required. The conformity standards against which the Bubble was tested prior to clinical use are detailed above, and we hope covered adequately in our revised section of the study methods, in response to your earlier suggestion.

- What are the lists of tasks/clinical duties gathered from the diary cards?

We gathered an extensive list of tasks from frontline staff who participated in the focus groups. During the clinical evaluation, we ensured that staff had undertaken all of these tasks whilst wearing Bubble-PAPR in the clinical environment. The list is long and we did not feel that it warranted inclusion in the paper. We propose to add the following text to the results section which we hope is satisfactory:

Fifteen staff contributed to the diary and focus group exercises, generating a list of tasks to be undertaken. One staff member from the 16 invited could not attend the focus group meeting. Staff reported a range of patient-facing activities, including: verbal communication between colleagues and patients; writing; typing; reading notes, computer screens and monitors; manual handling; invasive procedures; and maintenance of a clean/safe bedside environment. Over the course of the evaluation, staff completed all of the tasks identified by the diary exercise whilst wearing Bubble-PAPR in the clinical environment.

- What are the communication outcomes that were tested and how were they tested? The figure shows “I can be heard clearly by patients/colleagues”. Was this verified in any capacity? If not, please note this in the discussion as a limitation of the study as we cannot be sure that the other party was able to hear the wearer.

Thank you for this comment. We asked staff at the end of their stint wearing Bubble-PAPR simply the question that the reviewer has identified. We did not independently verify the communication was successful. Staff were however going about their usual work and we are confident that any limitations of two-way communication would have been recognised and reported.

Our study has some limitations. Some of the endpoints were self-reported by participating staff and not independently verified. This included communication between colleagues and between staff and patients. However, staff were performing their usual clinical duties whilst wearing Bubble-PAPR we are confident that any limitations of two-way communication would have been recognised and reported.

- Please report percentages of people who made certain categories of negative/positive comments. (i.e. 45% of people noted they had difficulty hearing wearing the PAPR, 65% commented they could see faces better, etc.). The word clouds do not give a quantitative summary of the results

Thank you for this suggestion. We have added the following text to the results section which provides the quantitative summary:

One hundred and thirty-two additional free text comments were reviewed and categorised into positive (n=47, 35.6%), negative (67, 50.8%) and neutral (18, 13.6%) comments (Figures S5-7 Supplemental). Most comments focused on the noise of the device, which improved throughout the project as the impellor and motor were made quieter in later design iterations. The categories and nature of comments were as follows: Noise (33 comments [3 neutral, 30 negative]); Comfort (24 comments [20 positive, 2 neutral, 2 negative]); Communication (22 comments [5 positive, 6 neutral, 11 negative]); General (21 comments [17 positive, 2 neutral, 2 negative]); Vision (14 comments [1 positive, 4 neutral, 9 negative]); Wear and fit (10 comments (2 positive, 1 neutral, 7 negative)); Stethoscope (5 negative comments); Safety (2 positive comments); Battery (1 negative comment).

- You mention that different iterations of the design made the impellor and motor quieter. Were safety and efficacy tests re-run on this iteration? Were any participants given the new iteration to wear with their daily activities and redo the questionnaire? If not, why? And please list a limitation in discussion.

The final iteration was tested by prior to use by a combination of INSPEC International and the MFT EBME team. The results section details that a further 22 staff evaluated this final version. Staff were from the same clinical locations and so the activity diary cards and focus groups were not repeated. Staff did complete the questionnaires, and these results are reported. Table 3 supplemental details the conformity testing that each iteration went through. We have only presented the fit-testing results from the first 10 participants (in table 5 supplemental). We have data for the first ten participants who used the final iteration and would be happy to include this if required. The Trial Safety Committee were reassured by the high fit-factors from the first round of safety tests which demonstrated, as expected for PAPRs, that the device performed its primary purpose of protecting staff from airborne particles. The overall fit-factors were at least ten times the required level for a 'pass'.

The third paragraph of the methods section should hopefully make the assessment process clear. Based on your earlier suggestions, this paragraph now reads:

Prior to first use, several device safety checks were independently undertaken by the MFT Electrical and Biomedical Engineering Department and INSPEC International Ltd, Salford, UK). A short report addressing the qualitative and quantitative criteria detailed in the relevant standards, and summarised in Table S2, S3 and S5 (Supplemental), was presented to the Trial Safety Committee.

Discussion

- Please make sure you publish data in your methods and results section to support your claim that the “Bubble-PAPR achieved its primary purpose of protecting staff from airborne potentially infectious material”. You must provide evidence.

We have presented the fit-test results that clearly show that the Bubble-PAPR protects staff from inward particulate matter. The fit-factors are at least 10 times the ‘pass’ level required. We believe that the opening paragraph of the discussion is clear and concise and does not make any claims that we do not provide supportive data for. We would be happy to re-word the second sentence of paragraph 1 of the discussion to include reference to the fit factors but feel that this would dilute the impact of the summary paragraph. This section is therefore unchanged for the resubmission.

- What are the “relevant simulated and clinical scenarios” that were determined by the diary cards? Were safety and efficacy tests done while performing these tasks? Why not?

Thank you for this relevant question. We have summarised the relevant scenarios identified from the diary cards in the section above, in response to a previous suggestion. The fit-testing procedure covers typical movements associated with these tasks and is sufficient for the relevant standards. From the results section:

Staff reported a range of patient-facing activities, including: verbal communication between colleagues and patients; writing; typing; reading notes, computer screens and monitors; manual handling; invasive procedures; and maintenance of a clean/safe bedside environment. Over the course of the evaluation, staff completed all of the tasks identified by the diary exercise whilst wearing Bubble-PAPR in the clinical environment.

- In the 3rd paragraph you say, “we adhered to relevant standards following a tiered evaluation within the governance structure”. Please detail what these standards were and how it was evaluated in the Methods and Results sections.

The standards we refer to are the conformity standards for PAPRs relevant at the time of development. These are detailed in the methods section and are (British Standard BS EN 12941 [Respiratory Protective Devices: Powered filtering devices incorporating a helmet or hood] and the European Union Personal Protective Equipment Directive EU2016/425). These standards are referenced. The results section details the assessment against these standards.

We have changed the text to hopefully make this clearer. This text now reads:

Despite the speed and agility demonstrated by the design team, we adhered to relevant conformity standards for PAPRs, following a tiered evaluation within the governance structure of an approved and regulated research project.

- Please explain or better support the sentence that starts, “this approach contrasted with many rapidly developed...”. It comes off as very dismissive without much specific evidence to support your claim. To what devices are you referring? What made them so reckless and inappropriate?

Why is yours so much better as you do not currently share any data demonstrating safety or efficacy of your design?

Thank you for this welcome challenge to the discussion text. We did not mean to come across as dismissive and value your feedback. The contrast that we are trying to highlight is that our device was developed rapidly in response to the pandemic, but were able to follow good design practices by establishing user needs, establish safety prior to use (fit testing using standard protocols), and confirm efficacy by collecting detailed user data. We have provided references for examples of devices to which we refer.

We have ‘toned down’ this section to now read:

This structured approach contrasted with some other rapidly developed or adopted pandemic RPE systems^{7, 13, 14}

- How is the Bubble PAPR different from other PAPRs?

- How does this PAPR’s design allow for these daily activities noted in the diary cards? What are the limitations of the design of this PAPR?

- Please expand upon the last sentence of the 4th paragraph. What specifically is unique about your study and how did you test it? What did the other studies not address?

We have addressed this group of comments together by amending the text of this paragraph. This now reads:

However, Bubble-PAPR was designed and developed to provide a viable alternative to FFP3 class face masks, in contrast to the more usual healthcare use of PAPRs. Other PAPRs are more complex, more cumbersome (belt-worn fans and hoses), more costly, and typically are selectively available on a limited basis to specific users or groups because of these factors. Our detailed analysis of work diary cards from various clinical staff ensured that Bubble-PAPR was used for all relevant procedures undertaken by medical, nursing, healthcare assistant, allied healthcare professional (speech and language therapy, physiotherapy, pharmacy), administrative and domestic staff in the clinical area. Staff were able to undertake their usual duties with this simple, collar-worn PAPR. Although the design is simple, with no electronic indicators or alarms, this did not impact on conformity testing or function. This perspective, testing safety, performance and the user experience, is unique within published respiratory protective equipment product evaluation studies.^{17, 18}

Conclusion

- First sentence: safety and efficacy must be demonstrated in this paper before this statement can be made.

We believe that our data clearly demonstrates that the Bubble-PAPR protects staff from the inhalation of airborne particulate matter, thus keeping them safe. We have not conducted any elaborate staff infection testing or surveillance as staff could contract respiratory illnesses from outside of the clinical areas in which Bubble-PAPR was used. We have attempted to qualify the statement further with the following addition:

Our study has demonstrated that Bubble-PAPR achieved its primary purpose of keeping staff safe from airborne particulate material whilst improving comfort, communication and the user experience when compared to usual RPE worn throughout the pandemic.

- If the design is patented, there should be no issue discussing the design and why it is superior and more effective than other designs. Please do so in the discussion. There also should be no problem publishing data to support its safety and efficacy.

As discussed above, we reference the patent which can be freely accessed for the interested reader. We have kept the design details brief for this clinical readership of this clinical journal.

- I recommend changing the last sentence. The last sentiment of your paper should not be putting other designs or studies down.

Thank you for this suggestion. On reflection, it appears that we are setting the wrong tone and we do not wish to appear to be putting down other researchers and developers who worked hard during the pandemic. We have therefore removed the final sentence.

Reviewer: 2: Dr. Akshay Kothakonda , Massachusetts Institute of Technology

Comments to the Author:

1. On Introduction section and Table 1, High Efficiency filter is defined as filtration efficiency of 99.95% of particulates smaller than 0.5 microns. What standard is this definition from? NIOSH defines HE filters as having 99.97% efficiency for particulate size of 0.3 micron.

Thank you for asking us to clarify the reference. The standard we used when making these calculations is EU Standard 149:2001 Respiratory Protective Devices. The definition of a P3 filter is filtering 99.95% of particles smaller than 0.5 micrometres. We have amended the title for the table to read:

Table 1. Classification of particulate filters, with a worked example and fit testing. Data from EU Standard 149:2001 Respiratory Protective Devices.

2. On table 1, calculation of APF for a P3 filter is not clear. If the filtration efficiency is 99.95%, PF should be 2000. How does this relate to APF of 20 for this filter? A more thorough definition of these terms may make it clearer. Also, a definition of qualitative fit factor it's distinction from PF and APF would be helpful to a reader, particularly in the discussion on the Methods section, relating nominal protection factor to applied fit factor.

We are sorry that we have not presented this confusing section with more clarity. We have amended the text in the table to be clearer about the different terminology and the worked example. The PF of 2000 that the reviewer has calculated is correct, but confusingly, this is the Nominal Protection Factor – a theoretical value, or one achieved in a sterile laboratory. The Assigned (or sometime 'Assumed') Protection Factor is a 'real world' value determined from testing the filter in the workplace environments for which it is intended to be used. Often the Assigned Protection Factor is a conservative metric, but this is the metric applied to the filters by the manufacturer, representing the minimum reduction in the ratio of outside:inside particulate concentration that can be expected.

Table 1. Classification of particulate filters, with a worked example and fit testing. Data from EU Standard 149:2001 Respiratory Protective Devices.

P1 – Filters about 80% of particles smaller than 2 micrometres

P2 – Filters about 94% of particles smaller than 0.5 micrometres

P3 – Filters about 99.95% of particles smaller than 0.5 micrometres

A respiratory protective device is considered adequate if it has the capacity to reduce the wearer's exposure to a hazardous substance to acceptable levels. The ratio of airborne particles outside:inside the filtering device gives a **nominal (theoretical) protection factor**. An **assigned protection factor** reflects the actual workplace conditions. For example: an airborne dust contaminant with an occupational exposure limit of 5mg/m³ may be present in the workplace in concentrations up to 60mg/m³ (determined by monitoring). A particle filter is needed to reduce the concentration by at least a factor of 12 (60/5=12). A P3 filter with an assigned protection factor of 20 would be suitable (as this is greater than the factor of 12 required). Other considerations such as exposure time, useability and disposal of the device need to be considered prior to undertaking a **fit test** with the intended wearer.

A fit test verifies that a **specific model** of device works as intended with a **particular individual**. For example, different face shapes and facial hair can interfere with a particular system's ability to filter environmental contaminants effectively.

Qualitative fit testing assesses the inward leakage past a mask of airborne compounds detectable by the wearer (typically bitter/sweet tasting substances), aerosolised using a spray device.

Quantitative fit testing measures particulate concentrations inside and outside of devices, typically undertaken by measuring sodium chloride aerosolised in water to generate a 'particle' count. Quantitative fit testing generates a **fit factor** – the ratio of airborne particle counts outside:inside. The fit factor takes account of the whole device (the filter, hood and airflow in the case of a PAPR). Fit factors for PAPRs are very high (optimal protection) and so, if correctly worn, fit testing prior to use is not usually required.

3. In the Introduction section, the authors correctly point out how many home-made PPE devices made during the pandemic lacked proper engineering design and safety evaluations. However, there have been some efforts, particularly from small industry and academia that have carried out PPE and PAPR development adhering to thorough safety design principles for these devices. For a complete background pertaining to this work, it might be worth highlighting some of these efforts in the Introduction section.

Thank you for this suggestion. In response to the comments from reviewer 1, we have made some changes to the introduction section. We have retained references 7, 8 and 9 which highlight efforts from small groups to developed RPE. We have also revised our conclusion so that we are not seen as being dismissive of these efforts.

4. One of the limitations of the current masks such as N95 masks for its use in clinical setting is the need for fit tests, as it has been pointed out in the paper. However, the subjects in the study also had to undergo qualitative fit tests prior to clinical use. This might be a limitation of the device. If the device is made so that no qualitative fit test is needed, then tests must be done (or results from tests already done) to show how many people failed fit tests. If these results don't exist, or if further tests cannot be undertaken, that should be noted as a limitation.

PAPRs do not require fit testing. This is because their fit factors are significantly in excess of FFP3 facemasks. We presented the fit testing data in order to demonstrate that our PAPR was safe, and that it achieved the required fit factor. We tested the first ten participants to use each iteration of the Bubble and presented the initial data in Table 5 supplemental in order to make the point that Bubble-PAPR was indeed safe and fit for purpose. We have added some additional text to the methods section which we hope clarifies this:

The first ten study participants to wear Bubble-PAPR underwent 'fit testing' with a particulometer (TSI Portacount Fit Tester 8040, TSI Instruments Ltd, Buckinghamshire, UK) following a standard protocol derived from the UK Government's Health and Safety Executive.¹² Fit testing is not required before wearing PAPRs, including Bubble-PAPR. The purpose of fit testing was to collect device performance data and to allow the research team to assure the Trial Safety Committee that Bubble-PAPR was performing to an appropriate standard.

5. In the "Strengths and limitations of this study" section, only strengths are mentioned. Please mention limitations as well.

Thank you for highlighting this omission. We have amended the strengths and limitations sections in the summary and conclusions. This was also highlighted by reviewer 1. The summary statements include:

- A limitation of our study was the design and evaluation were undertaken at a single (large) hospital, using similar staff groups (but different staff).

The discussion around limitations now reads:

Our study has some limitations. Some of the endpoints were self-reported by participating staff and not independently verified. This included communication between colleagues and between staff and patients. However, staff were performing their usual clinical duties whilst wearing Bubble-PAPR we are confident that any limitations of two-way communication would have been recognised and reported. The design of Bubble-PAPR addressed many of the issues identified by the same staff who subsequently evaluated the prototype. Whilst our study protocol allowed evaluation only within our Trust owing to the 'in-house' manufacturing exemption for testing, it is not unreasonable to expect similar results if our prototype were evaluated elsewhere. Although this may be considered a weakness of the study, many of the shortcomings of the PPE provided to frontline health workers around the world are well described and are essentially the same as those identified in our project.^{15, 16} Furthermore, we evaluated Bubble-PAPR against single-use and reusable FFP3 face masks, which could be construed as comparing two different classes of RPE. However, Bubble-PAPR was designed and developed to provide a viable alternative to FFP3 class face masks, in contrast to the more usual healthcare use of PAPRs. Other PAPRs are more complex, more cumbersome (belt-worn fans and hoses), more costly, and typically are selectively available on a limited basis to specific users or groups because of these factors. Although a pricing structure is currently unavailable, the simplicity of the design and components (designed with pandemic supply chain limitations in mind) means that Bubble-PAPR is likely to cost around 25-50% of the list price of equivalent PAPRs. Our detailed analysis of work diary cards from various clinical staff ensured that Bubble-PAPR was used for all relevant procedures undertaken by medical, nursing, healthcare assistant, allied healthcare professional (speech and language therapy, physiotherapy, pharmacy), administrative and domestic staff in the clinical area. Staff were able to undertake their usual duties with this simple, collar-worn PAPR. Although the design is simple, with no electronic indicators or alarms, this did not impact on conformity testing or function. This perspective, testing safety, performance and the user experience, is unique within published respiratory protective equipment product evaluation studies.^{17, 18}

Our study did not directly evaluate the patient experience with staff wearing different RPE by asking patients.

6. Abstract mentions that one of the objectives is to develop a low cost PAPR. Note the cost of Bubble PAPR so as to justify the accomplishment of this objective.

Thank you for raising this valid point. At the time of writing the paper, the Bubble was not commercially available. It has since been licensed by a major manufacturer but they haven't got a final cost. We do know that the ballpark figure will be about 25% of established competitor healthcare PAPRs. We have added the following statement to the discussion section:

Although a pricing structure is currently unavailable, the simplicity of the design and components (designed with pandemic supply chain limitations in mind) means that Bubble-PAPR is likely to cost around 25-50% of the list price of equivalent PAPRs.

VERSION 2 – REVIEW

REVIEWER	Gilbert, Catherine Tulane University School of Medicine
REVIEW RETURNED	16-Jan-2023

GENERAL COMMENTS	 • Abstract:  o Conclusion:  [ ] “Bubble-PAPR achieved its primary purpose ... the user experience.” Please add something along the lines of “as compared to a FFP3 face mask” [ ] “in contrast to many devices rapidly developed and deployed during the pandemic “ this comment is not particularly appropriate for your study. You compared your PAPR to FFP3 masks, not other PAPRs/devices. Additionally, you do not address other specific devices in your objectives, study design, or the rest of the paper. Your device has benefit and merit to the overall supply options, however, I do not think this comment is truly supported by the tests done or evidence provided in this study. You can say your device meets standards but unless you discuss other specific devices and how they do not meet standards and how that was harmful, it is inappropriate to keep this statement. • Strengths and Limitations of this study  o “undertaken at a single (large) hospital, using similar staff groups (but different staff)”  [ ] Recommend removing the parentheses around “large” and eliminating “but different staff”. • Introduction  o “respiratory protective equipment is used as part of a hierarchy of control measures and is usually considered a last resort”  [ ] How is RPE considered a last resort? I know you say it only “protects individual workers” but why does that make it a last resort? What other measures are you referring to? With respiratory diseases, masks are typically first line protection. o “false sense of security, encouraging risk-taking behaviours”  [ ] Like what? o “many restrict vision”  [ ] To be specific, they restrict visual field, not visual acuity. Unless you are referring to the fog that often occurs for people who wear glasses, but again that is more obstruction than altering of vision itself. o “none of these devices sought or achieved independent certification or provided data to support safety”  [ ] I would avoid absolutes. Please change “none” to “some” or “many” or “most” “did not seek or achieve...” • Methods  o I don’t see Table S5 in the supplemental documentation which is where some of the safety and efficacy testing is supposed to be displayed  o “Bubble-PAPR was then worn during simulated/clinical use where the usual tasks (identified in the focus groups) were undertaken”  [ ] I believe that you should put the detailed specifics (see below) of “patient facing activities” here in the methods rather than
---

the results. You can have them in both places but I think it should at least be in the methods section.

- verbal communication between colleagues and patients; writing; typing; reading notes, computer screens and monitors; manual handling; invasive procedures; emergency resuscitation; airway management; and maintenance of a clean/safe bedside environment

- Secondly: what were the job descriptions of the people who filled out the diary cards (and percentages)?

- Results

- o “Bubble-PAPR is shown schematically in Figure 1 (www.bubble-papr.com).”

- I have explored the website pretty thoroughly and I have only found the videos, not any figures. Please either make this more apparent/easier to find or put it in the supplemental files.

- o “Ninety-one staff wore Bubble-PAPR for a median of 45 (IQR 30-90, range 10-150) minutes” & “from a range of clinical and non-clinical roles”

- What was the breakdown of the positions held by these individuals? (% physicians, %nurses, %technicians, non-clinical, etc). This will be important to understand the background and primary job-practices of the individuals rating the device. Did people who had non-clinical rolls also participate in the simulation of invasive procedures, emergency resuscitation, and airway management? How many participants were clinical vs non clinical?

- o “During the initial phases, there was no significant difference between staff reporting ease of donning and doffing of Bubble-PAPR and usual PPE”

- By “usual PPE” to you mean the FFP masks? Please clarify.

- Discussion

- o “Bubble-PAPR achieved its primary purpose of protecting staff from airborne potentially infectious material”

- Please rephrase and say that it met standards for a PAPR device. “protecting staff from airborne” material is a little too generic of a claim for the scope of testing done.

- o “and communication with colleagues and patients (secondary outcomes) than usual RPE”

- Again, please specify FFP mask rather than “usual” RPE”

- o “However, staff were performing their usual clinical duties whilst wearing Bubble-PAPR we are confident that any limitations of two-way communication would have been recognised and reported”

- Can you say this with confidence? 11/91 (12%) people had negative comments about communication and 9/91 (10%) people had negative comments about vision. It is also unclear what specific clinical tasks people were performing. It makes sense that people did not have difficulty charting or taking a history or even placing an IV. But what about communication necessary running a code? If the Bubble-PAPR is geared toward health care workers, this should be addressed and mentioned.

- o “many of the shortcomings of the PPE provided to frontline health workers around the world are well described and are essentially the same as those identified in our project”

- Just because it is a shortcoming for others does not make it a non-issue. Consider rephrasing.

- o “Our detailed analysis of work diary cards from various clinical staff ensured that Bubble-PAPR was used for all relevant procedures undertaken by medical, nursing, healthcare assistant,

allied healthcare professional (speech and language therapy, physiotherapy, pharmacy), administrative and domestic staff in the clinical area”

I do not think you are able to make such a blanket statement. I have concerns on two fronts:

1) the safety aspect: As mentioned above, I think it is fair that you can make this statement for any clinical activity that fits into the test parameters you did: “normal breathing; deep breathing; turning head from side-to-side; moving head up and down; talking; bending over to 90 degrees; repeat normal breathing”. However, many clinical tasks require moving arms and shoulders. Your design is a rigid yoke that loosely rest on the shoulders. Therefore, any movement that involves movement of the clavicles or shoulders has the potential to alter the internal environment of the hood. The most concerning activities would be resuscitation (CPR, intubation), something that is critical for many healthcare workers, particularly in the setting of a severe respiratory disease. The Bubble-PAPR would not be secure and might very likely fall off while doing compressions. Even if the armpit straps were used (as I saw on some models), this would affect both comfort and would likely not maintain enough of a fit factor to keep safe. I have similar concerns for intubations where the arm wielding the scope is often at an odd angle that might compromise the respiratory environment of the wearer. I can also imagine scenarios where it might be compromised while placing an IV or many other procedures.

2) The bulkiness, looseness, and covering of the ears: How can you have multiple people effectively conducting a code wearing these? Ears are covered with an internal fan making communication difficult (you are rarely able to visualize the speaker in a code setting), multiple people leaning over a patient will bump PAPRs which will knock them loose compromising internal environments. You cannot use a stethoscope while using this device. This is a vital clinical task for patient assessment and procedures, particularly when talking about a respiratory illness.

These issues and limitation must be addressed explicitly in the limitation section if not throughout. This is another reason as to why it is necessary to talk about who filled out the task cards and exactly what clinical duties were performed by whom.

- Conclusion

- o “Our study has demonstrated that Bubble-PAPR achieved its primary purpose of keeping staff safe from airborne particulate material whilst improving comfort, communication and the user experience when compared to usual RPE worn throughout the pandemic”

Same concerns about broad statements rather than the specific variables tested during this study in addition to limitations of overall generalizability of healthcare worker rolls (physician, nurse, therapist, non-clinical, etc). Please modify to more accurately represent the scope of your study.

Please specify FFP3, not “usual RPE”

- Figures:

- o I do not believe Figures S5-7 should be included in the publication. The word clouds do not provide any real demonstration of data and they are confusing because the words that are large are not in the context in which they were said. For example “difficult” and “hear” are both large in the neutral and negative comments and it is impossible to know what was actually said or the message trying to be conveyed. Also if both neutral and negative both have the words large, how can one be considered

	negative and the other neutral? I think it best to just exclude these figures.
--	--

REVIEWER	Kothakonda , Akshay Massachusetts Institute of Technology, Aeronautics and Astronautics
REVIEW RETURNED	29-Dec-2022

GENERAL COMMENTS	All the comments have been addressed in this revision of the manuscript. I would ask the authors to consider expanding on the point noted in the Discussion section on design not having electronic indicators or alarms. Several markets around the world such as the US (through NIOSH) require PAPRs to have low flow indicators, which might limit the use of Bubble PAPR to American markets (or need additional design). This point may also be captured in the "Strengths and limitations of this study" section.
--

VERSION 2 – AUTHOR RESPONSE

Reviewer: 2

Dr. Akshay Kothakonda , Massachusetts Institute of Technology

Comments to the Author:

All the comments have been addressed in this revision of the manuscript.

I would ask the authors to consider expanding on the point noted in the Discussion section on design not having electronic indicators or alarms. Several markets around the world such as the US (through NIOSH) require PAPRs to have low flow indicators, which might limit the use of Bubble PAPR to American markets (or need additional design). This point may also be captured in the "Strengths and limitations of this study" section.

We are glad that we have addressed your previous comments and thank you for the time taken to re-review our manuscript.

Thank you for the suggestion around indicators and alarms. Our model did have mechanical/visual low flow indicators which were appropriate for the UK regulations at the time. We have addressed your comments however and this section now reads:

Although the design is simple, with visual/mechanical indicators instead of electronic indicators or alarms, this did not impact on conformity testing or function. Post-pandemic conformity requirements will vary around the world and future iterations of Bubble-PAPR may need to adapt to meet country-specific requirements. Addressing the actual activities undertaken by specific staff groups, testing safety, performance and the user experience, is unique within published respiratory protective equipment product evaluation studies.^{17, 18}

The word count does not permit us to add this as a specific 'weakness' in the bullet points, but we have included this text in the strength and weaknesses section of the discussion.

Reviewer: 1

Dr. Catherine Gilbert, Tulane University School of Medicine

Comments to the Author:

Dear Authors,

Thank you so much for all of your work and addressing the reviewer's concerns and insights. Please see attached notes for further comments and thoughts on the rewrite of your paper. References to your writing is in blue text and in quotes. My comments are below. Thank you for the honor of reviewing your work. [I have changed your text colour to purple for consistency with our responses in blue]

We are glad that we have addressed the majority of your previous comments and thank you for the time taken to re-review our manuscript.

ii Abstract:

o Conclusion:

- “Bubble-PAPR achieved its primary purpose ... the user experience.” Please add something along the lines of “as compared to a FFP3 face mask”
- “in contrast to many devices rapidly developed and deployed during the pandemic “ this comment is not particularly appropriate for your study. You compared your PAPR to FFP3 masks, not other PAPRs/devices. Additionally, you do not address other specific devices in your objectives, study design, or the rest of the paper. Your device has benefit and merit to the overall supply options, however, I do not think this comment is truly supported by the tests done or evidence provided in this study. You can say your device meets standards but unless you discuss other specific devices and how they do not meet standards and how that was harmful, it is inappropriate to keep this statement.

Thank you for these comments. This section of the abstract now reads:

Conclusions: Bubble-PAPR achieved its primary purpose of keeping staff safe from airborne particulate material whilst improving comfort and the user experience when compared with usual FFP3 masks. The design and development of Bubble-PAPR were conducted using a careful evaluation strategy addressing key regulatory and safety steps. ~~in contrast to many devices rapidly developed and deployed during the pandemic.~~

ii Strengths and Limitations of this study

- o “undertaken at a single (large) hospital, using similar staff groups (but different staff)”
 - Recommend removing the parentheses around “large” and eliminating “but different staff”.

Thank you for this comment. This section now reads:

- Limitations of our study include: design and evaluation undertaken at a single large hospital, using similar staff groups ~~(but different staff)~~; lack of formal independent cost analysis.

ii Introduction

- o “respiratory protective equipment is used as part of a hierarchy of control measures and is usually considered a last resort”

- How is RPE considered a last resort? I know you say it only “protects individual workers” but why does that make it a last resort? What other measures are you referring to? With respiratory diseases, masks are typically first line protection.
- o “false sense of security, encouraging risk-taking behaviours”
 - Like what?

Thank you for these two observations. We are limited in word count in the manuscript to address these interesting points in detail, but we have amended the first paragraph of the introduction to now read as follows (which we hope addresses your comments):

However, respiratory protective equipment is used as part of a hierarchy of control measures. ~~and is usually considered a last resort.~~ This is because RPE only protects individual workers, is prone to failure or misuse (wrong RPE for the wrong task/environment) and wearers may get a false sense of security, which may lead to neglect of other aspects of infection prevention and control, such as isolation requirements.³

- o “may restrict vision”
 - To be specific, they restrict visual field, not visual acuity. Unless you are referring to the fog that often occurs for people who wear glasses, but again that is more obstruction than altering of vision itself.

Thank you for this comment. Fair enough! We have changed this sentence to read:

...may restrict the visual field, limit communication,.....

- o “none of these devices sought or achieved independent certification or provided data to support safety”
 - I would avoid absolutes. Please change “none” to “some” or “many” or “most” “did not seek or achieve...”

Thank you for this comment. We have changed this sentence to read:

However, due to the urgency of the situation, few of these devices sought or achieved independent certification or provided data to support safety.⁸

ii Methods

- o I don't see Table S5 in the supplemental documentation which is where some of the safety and efficacy testing is supposed to be displayed

Table S5 appears in the uploaded files list of the author dashboard. We will check that it is incorporated into the final PDF this time.

- o “Bubble-PAPR was then worn during simulated/clinical use where the usual tasks (identified in the focus groups) were undertaken”
 - I believe that you should put the detailed specifics (see below) of “patient facing activities” here in the methods rather than the results. You can have them in both places but I think it should at least be in the methods section.
 - verbal communication between colleagues and patients; writing; typing; reading notes, computer screens and monitors; manual handling; invasive procedures; emergency resuscitation; airway management; and maintenance of a clean/safe bedside environment

Thank you for this comment. It was difficult to know where to report these activities, because at the time of setting the study up, we didn't know what they were, so they are actually results. However, we are happy to amend our methods. We have added the summary of the tasks to the methods to read:

Bubble-PAPR was then worn during simulated/clinical use where the usual tasks were undertaken (identified in the focus groups, including verbal communication between colleagues and patients; writing; typing; reading notes, computer screens and monitors; manual handling; invasive procedures; emergency resuscitation; airway management; and maintenance of a clean/safe bedside environment).

i Results

- Secondly: what were the job descriptions of the people who filled out the diary cards (and percentages)?

We have added the following detail to the results section:

Fifteen staff contributed to the diary and focus group exercises. Nurses (n=7), Doctors (4) Physiotherapists (2), Advanced Practitioners (1), Speech and Language Therapists (1) representing Emergency Medicine, Critical Care, Orthopaedics and Obstetric specialties generated a list of tasks to be undertaken.

- o “Bubble-PAPR is shown schematically in Figure 1 (www.bubble-papr.com).”
 - I have explored the website pretty thoroughly and I have only found the videos, not any figures. Please either make this more apparent/easier to find or put it in the supplemental files.

We thank the reviewer for her detailed checking of our references. We felt that the animations were adequate for the likely readership of the paper. There are detailed diagrams available in the publicly available patent, available from www.espacenet.com. We have added this signpost to the section you describe:

The final design of Bubble-PAPR is shown schematically in Figure 1 (www.bubble-papr.com, with detailed technical drawings available by searching the patent number [PCT/GB2021/052147] at www.espacenet.com

- “Ninety-one staff wore Bubble-PAPR for a median of 45 (IQR 30-90, range 10-150) minutes” & “from a range of clinical and non-clinical roles”
 - What was the breakdown of the positions held by these individuals? (% physicians, %nurses, %technicians, non-clinical, etc). This will be important to understand the background and primary job-practices of the individuals rating the device. Did people who had non-clinical rolls also participate in the simulation of invasive procedures, emergency resuscitation, and airway management? How many participants were clinical vs non clinical?

The breakdown of staff roles is described in Figure 2 Supplemental:

- “During the initial phases, there was no significant difference between staff reporting ease of donning and doffing of Bubble-PAPR and usual PPE”
 - By “usual PPE” to you mean the FFP masks? Please clarify.
 -

Yes, we meant usual FFP3 masks. We have amended this section to add clarity as you suggest:

During the initial phases, there was no significant difference between staff reporting ease of donning and doffing of Bubble-PAPR and usual PPE (the FFP3 face masks which staff had used for many months at the time of the evaluation).

i Discussion

- “Bubble-PAPR achieved its primary purpose of protecting staff from airborne potentially infectious material”
 - Please rephrase and say that it met standards for a PAPR device. “protecting staff from airborne” material is a little too generic of a claim for the scope of testing done.

- “and communication with colleagues and patients (secondary outcomes) than usual RPE”
 - Again, please specify FFP mask rather than “usual” RPE”

Thank you for these suggestions. We have tidied up this section to now read:

Bubble-PAPR achieved its primary purpose of protecting staff by exceeding recognised safety standards for PAPRs, ~~from airborne potentially infectious material~~ whilst also being rated significantly higher for comfort (the primary outcome), perceived safety, and communication with colleagues and patients (secondary outcomes) than usual FFP3 face masks.

- “However, staff were performing their usual clinical duties whilst wearing Bubble-PAPR we are confident that any limitations of two-way communication would have been recognised and reported”
 - Can you say this with confidence? 11/91 (12%) people had negative comments about communication and 9/91 (10%) people had negative comments about vision. It is also unclear what specific clinical tasks people were performing. It makes sense that people did not have difficulty charting or taking a history or even placing an IV. But what about communication necessary running a code? If the Bubble-PAPR is geared toward health care workers, this should be addressed and mentioned.

Thank you for highlighting this. We have amended this section to now read:

However, staff were performing their usual clinical duties whilst wearing Bubble-PAPR ~~we are confident that~~ and any limitations of two-way communication ~~would have been~~ recognised and reported.

- “many of the shortcomings of the PPE provided to frontline health workers around the world are well described and are essentially the same as those identified in our project”
 - Just because it is a shortcoming for others does not make it a non-issue. Consider rephrasing.

We have reflected on this phrase and think that it is accurate and appropriate, and referenced. We have elected to leave this statement unchanged.

- “Our detailed analysis of work diary cards from various clinical staff ensured that Bubble-PAPR was used for all relevant procedures undertaken by medical, nursing, healthcare assistant, allied healthcare professional (speech and language therapy, physiotherapy, pharmacy), administrative and domestic staff in the clinical area”
 - I do not think you are able to make such a blanket statement. I have concerns on two fronts:

Thank you for this observation. We have adapted our statement to be slightly less 'blanket'. It now reads:

Our detailed analysis of work diary cards from various clinical staff ensured that Bubble-PAPR was used for all relevant procedures identified by participating staff in our settings that were undertaken by medical, nursing, healthcare assistant, allied healthcare professional (speech and language therapy, physiotherapy, pharmacy), administrative and domestic staff in the clinical area

i Conclusion

- 1) the safety aspect: As mentioned above, I think it is fair that you can make this statement for any clinical activity that fits into the test parameters you did: "normal breathing; deep breathing; turning head from side-to-side; moving head up and down; talking; bending over to 90 degrees; repeat normal breathing". However, many clinical tasks require moving arms and shoulders. Your design is a rigid yoke that loosely rest on the shoulders. Therefore, any movement that involves movement of the clavicles or shoulders has the potential to alter the internal environment of the hood. The most concerning activities would be resuscitation (CPR, intubation), something that is critical for many healthcare workers, particularly in the setting of a severe respiratory disease. The Bubble-PAPR would not be secure and might very likely fall off while doing compressions. Even if the armpit straps were used (as I saw on some models), this would affect both comfort and would likely not maintain enough of a fit factor to keep safe. I have similar concerns for intubations where the arm wielding the scope is often at an odd angle that might compromise the respiratory environment of the wearer. I can also imagine scenarios where it might be compromised while placing an IV or many other procedures.
- 2) The bulkiness, looseness, and covering of the ears: How can you have multiple people effectively conducting a code wearing these? Ears are covered with an internal fan making communication difficult (you are rarely able to visualize the speaker in a code setting), multiple people leaning over a patient will bump PAPRs which will knock them loose compromising internal environments. You cannot use a stethoscope while using this device. This is a vital clinical task for patient assessment and procedures, particularly when talking about a respiratory illness.
- These issues and limitation must be addressed explicitly in the limitation section if not throughout. This is another reason as to why it is necessary to talk about who filled out the task cards and exactly what clinical duties were performed by whom.

Thank you for these relevant comments. During the sim-suite testing of the Bubble, we ran through multiple high-stakes simulated events, including complex airway interventions, cardiac arrests and management of the critically ill patient (anaphylaxis, septic shock, for example) in order to evaluate the PAPR in the situations that we might find during the clinical tests. There is video footage of on the website if some of these tests including CPR and intubation, showing that 1) the collar doesn't fall off (one of our original concerns) and 2) communication was perfectly possible in these high stress cases. The ability to see each other's faces and communicate using non-verbal facial expressions was highlighted by staff as particularly important.

You are correct that the current model means you cannot use a stethoscope. We were able to get a Bluetooth stethoscope to work with the system but this is not suitable or available for many practitioners. I would argue that it isn't a "vital clinical task" in many circumstances, particularly in the ICU where there is an increasing recognition of assessment modalities such as lung/cardiac ultrasound. I don't recall using a stethoscope at all during the pandemic myself, and I was the clinical lead on a large ICU most days. We recognise that this may not be a universal position however and so have adapted the text as follows to address your comments:

In methods:

In order to evaluate critical communication and the stability of the Bubble-PAPR, the simulated environment tests also included high-stakes team-based tasks such as managing a cardio-respiratory arrest, cardiopulmonary resuscitation, assessment and management of the critically ill patient and complex airway management.

In the limitations paragraph of the discussion:

Limitations of the design include the inability to use a conventional stethoscope (although Bluetooth stethoscopes were used effectively), potential visual distortions if the visor section of the hood became creased, and the residual noise during use (common amongst PAPRs).

- “Our study has demonstrated that Bubble-PAPR achieved its primary purpose of keeping staff safe from airborne particulate material whilst improving comfort, communication and the user experience when compared to usual RPE worn throughout the pandemic”
 - Same concerns about broad statements rather than the specific variables tested during this study in addition to limitations of overall generalizability of healthcare worker rolls (physician, nurse, therapist, non-clinical, etc). Please modify to more accurately represent the scope of your study.
 - Please specify FFP3, not “usual RPE”

Thank you for this suggestion. We have amended this paragraph to match the earlier suggestions from yourself and the editor in the abstract. This section of the conclusion now reads:

Our study has demonstrated that Bubble-PAPR achieved its primary purpose of keeping staff safe from airborne particulate material whilst improving comfort, communication and the user experience when compared with usual FFP3 face masks worn throughout the pandemic.

ii Figures:

- I do not believe Figures S5-7 should be included in the publication. The word clouds do not provide any real demonstration of data and they are confusing because the words that are large are not in the context in which they were said. For example “difficult” and “hear” are both large in the neutral and negative comments and it is impossible to know what was actually said or the message trying to be conveyed. Also if both neutral and negative both have the words large, how can one be considered negative and the other

tral? I think it best to just exclude these figures.

We have had quite positive feedback from others about the word clouds. The authors quite liked them and we felt it made it clear that we had collected and considered all of the free text responses. We agree that the word clouds do not report the full text but we felt that they gave a flavour of what was reported without another dense section of text.

We concluded that we would like to leave these figures in but would accept the decision of the editors if the BMJ-Open team thought that these were best removed.

Thank you for your detailed review of our work.

VERSION 3 - REVIEW

REVIEWER	Gilbert, Catherine Tulane University School of Medicine
REVIEW RETURNED	16-Mar-2023

GENERAL COMMENTS	 • Abstract:  o “We hypothesised that participants would rate Bubble-PAPR more highly...” [ ] Please specify that you mean it would be rated more highly in the “domains of comfort, perceived safety and communication.” (-from the end of your introduction) • Strengths and Limitations of this study  o Bubble-PAPR is an excellent example of developing a cosmopolitan network (social networks across historical, political, and cultural boundaries). These networks could become a key feature of future system resilience [ ] This is an interesting point and I would be very curious to hear more. I see two short sentences in the conclusion, but if it is going to be considered a strength of the study I would recommend spelling it out a little more, maybe in the discussion. • Introduction – looks great • Methods – looks great o “A short report addressing the qualitative and qualitative criteria detailed in the relevant standards...” [ ] I think you may mean qualitative and quantitative • Results – looks great • Discussion  o “Other PAPRs are more complex, more cumbersome (belt-worn fans and hoses), more costly, and typically are selectively available on a limited basis to specific users or groups because of these factors.” [ ] Do you have a source for this? o After greater thought, my previous comment that discussed concerns about the Buble-PAPR during resuscitation (CPR, intubation, other more involved procedures) is more about the safety and efficacy of the device in these situations, not comfort or perceived safety, and is therefore outside the scope of the direct objectives and procedures of this paper. However, from an engineering perspective it might be something to consider that, because you have modified the design of the PAPR (changing the position of the fan, how the hood secures, etc.), your design may need to meet different standards as the design changes how it might function in different settings/activities. I would like to see it as a possible limitation in your discussion to consider.
--

	 • Conclusion – looks good • Figures:  o I still feel that Figures S5-7 (word clouds) only provide an ambiguous qualitative demonstration of subjective data. It makes me concerned that this would set a poor precedent for data presentation. You have done an excellent job of writing out the data in percentage for in the results section. Perhaps a bar graph would be a better and more accurate representation?
--	--

VERSION 3 – AUTHOR RESPONSE

Reviewer 1. Dr. Catherine Gilbert, Tulane University School of Medicine

Thank you so much for taking my thoughts and comments into consideration. I really appreciate your time and the work you have done on both the device and the paper. I have just a few more comments I think would enhance the paper. Please see attached. Thank you so much.

Abstract:

o “We hypothesised that participants would rate Bubble-PAPR more highly...”

Please specify that you mean it would be rated more highly in the “domains of comfort, perceived safety and communication.” (-from the end of your introduction)

We have changed the abstract text to now read:

We hypothesised that participants would rate Bubble-PAPR more highly than current FFP3 face mask respiratory protective equipment (RPE) in the domains of comfort, perceived safety and communication.

Strengths and Limitations of this study

o Bubble-PAPR is an excellent example of developing a cosmopolitan network (social networks across historical, political, and cultural boundaries). These networks could become a key feature of future system resilience

This is an interesting point and I would be very curious to hear more. I see two short sentences in the conclusion, but if it is going to be considered a strength of the study I would recommend spelling it out a little more, maybe in the discussion.

Thanks. We have added a little more text to the discussion which hopefully explains what we mean. The discussion text now reads:

We recommend others to follow the framework proposed by Duggan et al. for the development of novel medical devices, with regular reviews of safety and useability data within the framework of a robust and transparent clinical trial.⁷ The development of Bubble-PAPR required the rapid formation of a cosmopolitan network of frontline healthcare staff, designers, engineers, academics, innovators, marketing experts, manufacturers and funders. Our collaborative had not all worked together before and members crossed historical, political, and cultural boundaries to work effectively together. Post-pandemic, cosmopolitan networks such as this could become a key feature of future system resilience and facilitate new ways of working.

We have added the following text to the final conclusion to follow on from the other editorial comment, and changes to the strengths/weaknesses bullet points:

The development of Bubble-PAPR is an excellent example of growing a cosmopolitan network across historical, political, and cultural boundaries that could become a key feature of future system resilience.

Introduction – looks great

Methods – looks great

o “A short report addressing the qualitative and quantitative criteria detailed in the relevant standards...”

I think you may mean qualitative and quantitative

Yes – thanks for spotting our mistake.

Results – looks great

Discussion

o “Other PAPRs are more complex, more cumbersome (belt-worn fans and hoses), more costly, and typically are selectively available on a limited basis to specific users or groups because of these factors.”

Do you have a source for this?

It is difficult quote a source for this. There isn't a comprehensive review in the literature that we found that discusses different PAPRs. We could add an example website (such as https://www.3m.com/3M/en_US/respiratory-protection-us/products/papr/) but that rather singles out a particular company. Listing all the major companies then excludes the smaller ones, so we think it is best just to leave this as a statement of fact.

o After greater thought, my previous comment that discussed concerns about the Bubble-PAPR during resuscitation (CPR, intubation, other more involved procedures) is more about the safety and efficacy of the device in these situations, not comfort or perceived safety, and is therefore outside the scope of the direct objectives and procedures of this paper. However, from an engineering perspective it might be something to consider that, because you have modified the design of the PAPR (changing the position of the fan, how the hood secures, etc.), your design may need to meet different standards as the design changes how it might function in different settings/activities. I would like to see it as a possible limitation in your discussion to consider.

OK. This is a fair point. We have amended the strengths/weaknesses discussion section to now read:

Addressing the actual activities undertaken by specific staff groups, testing safety, performance and the user experience, is unique within published respiratory protective equipment product evaluation studies.^{17, 18} High acuity activities such as CPR and tracheal intubation were undertaken whilst wearing Bubble-PAPR but we collected data only around perceived comfort, safety and self-reported efficacy. Bubble-PAPR meets current industrial standards for the safe use of respiratory protection, but such standards are not usually designed with healthcare procedures in mind. Post-pandemic conformity requirements will vary around the world and future iterations of Bubble-PAPR may need to adapt to meet country-specific requirements.

Conclusion – looks good

Figures:

o I still feel that Figures S5-7 (word clouds) only provide an ambiguous qualitative demonstration of subjective data. It makes me concerned that this would set a poor precedent for data presentation. You have done an excellent job of writing out the data in percentage for in the results section. Perhaps a bar graph would be a better and more accurate representation?

Thanks for this comment. We still liked the clouds as a simple presentation of the data we collected. We had intended to give the editorial team free reign on whether to include these or not, but the authors do not feel strongly about their inclusion or not. Splitting them into sections and themes creates un-readable tables and we don't think this adds much to the paper.

Thanks you for your constructive and detailed reviews which we think have significantly enhanced the paper from its original state.